# Magnesium Plus Hydrogen Fertilization Enhances Mg Uptake, Growth Performance and Monoterpenoid Indole Alkaloid Biosynthesis in *Catharanthus roseus*

**DOI:** 10.3390/plants14213336

**Published:** 2025-10-31

**Authors:** Chaoqin Ou, Xinmin Qiu, Qian Zhao, Ding Ding, Yaojie Zhang, Ji De, Yuliang Wang, Kexuan Tang, Haiyan Yang, Qifang Pan

**Affiliations:** 1Frontiers Science Center for Transformative Molecules, Joint International Research Laboratory of Metabolic & Developmental Sciences, Plant Biotechnology Research Center, Fudan-SJTU-Nottingham Plant Biotechnology R&D Center, School of Agriculture and Biology, Shanghai Jiao Tong University, Shanghai 200240, China; ocq.127@sjtu.edu.cn (C.O.); seem-ing-1110@sjtu.edu.cn (X.Q.); zhaoqianbh@sjtu.edu.cn (Q.Z.); ding798@sjtu.edu.cn (D.D.); shalom@sjtu.edu.cn (Y.Z.); wangyuliang@sjtu.edu.cn (Y.W.); 2Key Laboratory of Biodiversity and Environment on the Qinghai-Tibetan Plateau, Ministry of Education, School of Ecology and Environment, Tibet University, Lhasa 850000, China; dekyi1981@utibet.edu.cn; 3Yazhouwan National Laboratory, 8 Huanjin Road, Sanya 572025, China; tangkexuan@yzwlab.cn; 4Shanghai Key Laboratory of Hydrogen Science, Center of Hydrogen Science, Shanghai Jiao Tong University, Shanghai 200240, China

**Keywords:** *Catharanthus roseus*, magnesium plus hydrogen fertilization, Soil pH, Mg uptake, plant growth, monoterpenoid indole alkaloids

## Abstract

Magnesium and hydrogen fertilization have been reported to contribute to plant growth and metabolite production. Simultaneous fertilization of magnesium and hydrogen is a promising strategy for plant development and secondary metabolism, but remains unexplored in *Catharanthus roseus* (L.) G.Don, which produces varieties of monoterpenoid indole alkaloids (MIAs). This study conducted a glasshouse experiment comprising five treatments: MgO, MgSO_4_, MgH_2_, magnesium powder (MgP), and the control, to investigate and compare the effects of Mg fertilizers and Mg plus H_2_ fertilizers on soil pH, Mg uptake, seed germination, plant growth, and MIA biosynthesis in *C. roseus*. Application of MgH_2_, MgP, and MgO fertilizers significantly raised soil pH to 6.14~6.38. MgH_2_ and MgP fertilization significantly increased plant weight by 60% and 29% over the control, respectively. MgH_2_ and MgP produced greater increases in Mg content, chlorophyll content, plant height, and weight than MgO and MgSO_4_. Four fertilizers up-regulated the expression of most MIA biosynthetic genes, especially those in the vindoline pathway. Among them, MgH_2_ yielded the highest contents of catharanthine, vindoline, and ajmalicine, reaching 167%, 149% and 517% of the control, respectively. Pearson correlation analysis showed significant positive correlations among H_2_ release, soil pH, and Mg uptake, as well as with plant growth and MIA content. These findings suggest that Mg plus H_2_ fertilizers released H_2_ and increased soil pH to promote Mg uptake, chlorophyll contents, plant growth, and MIA production in *C. roseus*, highlighting the potential of MgH_2_ and Mg powder as innovative fertilizers to enhance alkaloid yields in medicinal plants.

## 1. Introduction

*Catharanthus roseus* (L.) G. Don is a member of the *Apocynaceae* family and a rich repository of over 130 monoterpenoid indole alkaloids (MIAs), which are known for their significant pharmacological activities [1]. Notably, vinblastine and vincristine, two dimeric indole alkaloids derived from this plant, are well known for their anti-cancer effects [2]. Besides, *C. roseus* also produces monomeric MIAs, such as ajmalicine, used as an antihypertensive, and serpentine, used as a sedative [1]. The small amounts of vinblastine and vincristine in *C. roseus* plants make the semi-synthetic route from the monomeric precursors (vindoline and catharanthine) the main commercial production of the two dimeric MIAs. Therefore, optimizing cultivation strategies is critical to increase the availability of pharmacologically valuable MIAs in *C. roseus* plants.

Fertilization is a common way to ensure the yield of crops. The application of fertilizers with macronutrients or micronutrients has been reported to impact the accumulation of alkaloids in *C. roseus*. Different nitrogen forms applied had different influences on the MIA production in *C. roseus* [3]. The simultaneous supply of higher nitrate nutrition under UV-B radiation resulted in an increase of catharanthine and vindoline accumulation, but UV-B stress obviously inhibited the growth of *C. roseus* [4]. Higher potassium concentration applied could effectively regulate MIA metabolism, but reduced the biomass of *C. roseus* [5]. Urea fertilizer and substrates enhanced the production of reserpine in *C. roseus* plants [6]. Nitrogen fertilization significantly increased the alkaloid content in *C. roseus* mutants [7]. Exogenous CaCl_2_ could promote the content of vindoline, catharanthine, and vinblastine under salt stress [8]. Nano-iron fertilizers increased growth parameters, photosynthetic pigments, and total protein contents in the treated *C. roseus* significantly [9]. Foliar application of zinc (Zn, 100 ppm) was found to be effective for enhancing the vindoline content [10]. In *C. roseus*, deficiencies in magnesium (Mg) reduced ajmalicine concentration by 22% when compared with complete treatment [11]. Taken together, these studies indicate that nutrient supply can modulate MIA biosynthesis, although often at the expense of growth performance. Despite extensive studies on N, K, Ca, and Zn, few have investigated Mg application, and none have explored the combined role of Mg and H_2_. This undiscovered territory constitutes the primary novelty of our work. 

Magnesium (Mg) is an essential macronutrient for plants and is involved in photosynthesis, assimilate production, and distribution [12]. Mg facilitates the absorption and utilization of N, and the application of Mg fertilizer improves the grain yield and NUE of plants [13,14,15]. Besides, Mg fertilizer also affected the alkaloid metabolism in plants. Both Mg fertilizers (MgSO_4_ and CaMg(CO_3_)_2_) increased the glycoalkaloid (TGA) content of Katahdin potato tubers [16]. Glycoalkaloid accumulation approached maximum levels in potato tubers fertilized at a rate of 40 lb of MgS0_4_/acre [17]. MgS0_4_ leaf fertilizer (2% solution) had a positive effect on the formation of total alkaloid content in the Indian tobacco, too [18]. Silicon-calcium-potassium-magnesium fertilizer results in higher levels of alkaloid, polysaccharide, flavone, and total protein in *Codonopsis tangshen* Oliv compared to the controls [19]. Mg fertilizers were classified into two types: (1) slowly released (Mg-S) fertilizers, including MgO and Mg(OH)_2_, and (2) rapidly released (Mg-R) fertilizers, including MgSO_4_ [20]. These findings suggest that Mg fertilization can influence alkaloid metabolism in various plant species. It was reported that Mg deficiencies reduced ajmalicine concentration in *C. roseus* [11], yet its role in MIA production in *C. roseus* remains unclear.

Hydrogen has been found to differentially promote the germination of seeds in rye, mung bean, and rice [21]. Furthermore, hydrogen-rich water has been reported to influence plant hormone synthesis, upregulate the expression of certain genes, and interact with signaling molecules, thereby regulating plant growth and development [22]. Critically, emerging evidence suggests that molecular hydrogen can directly regulate the biosynthesis of secondary metabolites. For instance, treatment with hydrogen-rich water (HRW) is documented to significantly upregulate the phenylpropanoid biosynthesis pathway and increase the content of flavonoids and coumarins in the medicinal plant *Ficus hirta* [23]. Magnesium hydride (MgH_2_) is a solid-state hydrogen source with high hydrogen-storage capacity (7.6 wt%), low cost, and abundant resources, and can be potentially applied in industry, medicine, and agriculture [24,25,26,27]. Upon hydrolysis, one molecule of MgH_2_ releases two molecules of H_2_. Mg powder (MgP) has a similar reaction in water, but produces one molecule of H_2_. Their ability to release H_2_ enables both MgH_2_ and MgP to act as combined Mg and H_2_ fertilizers. Furthermore, recent evidence has revealed that the Mg-H_2_ combination, supplied as MgH_2_, can activate complex defense mechanisms in plants, as demonstrated by its ability to enhance cadmium tolerance in rice through hydrogen-mediated regulation [28]. However, the potential of this synergy to influence the biosynthesis of valuable secondary metabolites in medicinal plants remains unexplored.

In this study, four fertilizers, MgO, MgSO_4_, MgH_2_, and MgP with the same Mg concentration, are applied to investigate the effect of magnesium, or plus hydrogen fertilization on soil pH, Mg uptake, seeds germination, growth, MIA biosynthesis and gene expression of *C. roseus* plants, exploring the potential use of MgH_2_ and Mg powder as fertilizers in agriculture. We hypothesized that Mg H_2_ and Mg powder, by simultaneously supplying Mg and releasing H_2_, would improve Mg uptake, plant growth, and MIA biosynthesis in *C. roseus* compared to conventional Mg fertilizers. Given the limited information on hydrogen’s effects on secondary metabolites and the lack of studies on combined Mg+H_2_ application in medicinal plants, it is important to evaluate their impact on *C. roseus* growth and MIA production.

## 2. Materials and Methods

### 2.1. Plant Materials and Treatments

Seeds of *C. roseus* (Pacific Cherry Red cultivar) were purchased from Limei Horticulture Technology Co., Ltd. (Suzhou, China), and germinated in the substrate soils treated with four forms of magnesium fertilizers, MgSO_4_, MgO, MgH_2_, and Mg powder (MgP) (50 μm, 98%), representing conventional Mg rapid- or slow-release sources, and novel Mg + H_2_ slow-release sources. Four treatments with different forms of magnesium fertilizers were defined as having the same magnesium concentration of 7 mg/kg soil. After germination for 2 weeks, seedlings were transplanted from trays into pots (1 plantlet per pot), and the same four Mg fertilizer treatments were again applied to the soil. Four weeks after the second Mg fertilization, twenty plantlets were collected for each treatment. After the measurement of plant height and weight, leaves from the top three layers of *C. roseus* plantlets were ground into powder in liquid nitrogen and stored in a −80 °C freezer for the following analysis. Samples for MIA measurement were lyophilized for 72 h.

The experiment was conducted in the greenhouse at 25 ± 2 °C. The substrate soil was made of peat soil and vermiculite at a volume ratio of 6:4. MgSO_4_·7H_2_O was purchased from Jiuding Chemical (Shanghai, China) Technology Co., Ltd. MgO was purchased from Meryer Experimental Equipment (Shanghai, China) Co., Ltd. MgH_2_ and Mg were provided by the Center of Hydrogen Science, Shanghai Jiao Tong University. The control treatment was applied without any magnesium fertilizers.

### 2.2. Seedling Emergence

Plastic trays were filled with an 8-cm layer of the substrate soil and treated with four different magnesium fertilizers as above. The test was made with three replications of 30 seeds per tray for each treatment. Seeds were buried 1 cm deep, covered with substrate, and irrigated once a week with 2 L of water. Emerged seedlings were counted at 10 pm daily. They were assumed to emerge when the cotyledons were out of the soil.

After sowing, data was taken over 14 days. The percentage of emergence and the time for emergence of 50% of the seeds were calculated. The time for 50% emergence of *C. roseus* seedlings was calculated by interpolation in order to include the fraction of the day. The equation was T50% = T + (50% − X)/ (Y − X), in which T50% is the time for emergence of 50% of the seeds, T is the day before 50% was reached, X is the emergence (%) observed in T, and Y is the emergence (%) in the day it was ≥50%.

### 2.3. Determination of Plant Height and Weight

Four weeks after the second Mg fertilization, five plantlets per replicate, each Mg fertilizer treatment was randomly selected for measuring plant height and weight (fresh weight, FW). The plant height was measured using a ruler (accuracy of 0.5 mm), and the weight was measured in a pool per replicate using an analytical balance (accuracy of 0.0001 g).

### 2.4. ChlorophyII Analysis

*C. roseus* leaf samples were flash frozen and ground into powder in liquid nitrogen. 0.1 g sample powder was mixed in 1 mL 80% acetone buffer and stayed for 5 min in the dark. Extracts were centrifuged at 12,000 r/min for 10 min. Supernatant was transferred to a 2.0 mL EP tube. Pellets were resuspended twice in 1 mL 80% acetone buffer, once more for the supernatant. Finally, the supernatant was filtered by an organic needle filter, and 200 μL was taken into a microplate for the detection of the microplate scanning spectrophotometer (PowerWave XS, BIO-TEK instruments, Inc., Winooski, VT, USA). The absorbance of the supernatant was read at 663 and 645 nm against a blank containing 80% acetone buffer. Each treatment was replicated three times. The concentration of chlorophylls a and b in acetone extracts was then calculated using the Arnon equations as follows:ρ_a_ =12.72A_663_ − 2.59A_645_(1)ρ_b_ =22.88A_645_ − 4.67A_663_(2)

### 2.5. Determination of Mg Content of Plant Tissue

Accurately weigh 0.1 g of the sample into a 50 mL centrifuge tube, precise to 0.0001 g. Add 5 mL of water to moisten, then add 2 mL of nitric acid and 1 mL of hydrogen peroxide. Heat at 80 °C for 30 min, followed by heating at 125 °C for 2 h. After cooling to room temperature, transfer the solution to a 50 mL volumetric flask and dilute to the mark with ultrapure water, then mix well. After the pretreatment, determine the magnesium content in the plant leaves using Inductively Coupled Plasma-Optical Emission Spectrometry (ICP-OES, iCAP 7000, Thermo Fisher, Waltham, MA, USA). The digestion vessel type was teflon, and blanks and standards were included for ICP-OES calibration.

### 2.6. Relative Expression Analysis via Quantitative RT-PCR

For quantitative RT-PCR (qRT-PCR), total RNA was isolated from the leaves stored at −80 °C. DNA contamination was removed using DNase I following the protocol provided by the manufacturer (TaKaRa, Osaka, Japan). The cDNAs were synthesized from 500 ng RNA samples using Prime ScriptTM Reverse Transcriptase Reagent according to the manufacturer’s instructions, using oligo (dT) as primer. The qRT-PCR analysis was performed in Peltier Thermal Cycler PTC200 (Bio-Rad), using the cDNAs as templates and gene-specific primers (Appendix A) for the gene analysis. The primers for the MIA biosynthetic genes are listed in Appendix A. SYBR Green (SYBR Premix Ex Taq; TaKaRa) was used in the PCR reactions to quantify the amount of dsDNA. qRT-PCR cycling conditions were listed in Appendix A. The relative Ct (threshold cycle value) method (User Bulletin 2, ABIPRISM700 Sequence Detection System, update 2001; PerkinElmer/Applied Biosystems) was used to estimate the initial amount of template present in the reactions.

### 2.7. MIAs Analysis

UPLC-Q/TOF MS analyses were performed using a Primer UPLC-Q-TOF mass spectrometer (Waters Corp., Milford, MA, USA) equipped with an electrospray ionization source. Data acquisition, handling, and instrument control were performed using MassLynx 4.1 software. Mass range, *m*/*z* 50 to 1000 in positive mode; capillary, 3.0 kV; sample cone voltage, 35 V; extraction cone, 3.0; ion guide, 3.0; source temperature, 115 °C; desolvation gas temperature, 300 °C; flow rate of desolvation gas, 700 L h^−1^.

To ensure accuracy and reproducibility, all analyses were conducted using an independent reference spray via the Lock Spray interface; Tyr–Gly–Gly–Phe–Leu (leucine-encephalin, 200 pg μL^−1^) was used as a lock mass (*m*/*z* 556.2771) under positive-ion conditions for real-time calibration (flow rate of 30 μL min^−1^). Before the experiment, a single-point calibration was performed against the lock mass compound (leucine-encephalin). A multiple-point calibration was then performed over the range *m*/*z* 50–1000 using sodium formate solution (prepared from 10% formic acid/0.1 mol L^−1^ sodium hydroxide solution/acetonitrile, 10 mL/10 mL/80 mL). All points fell within 1 ppm during calibration. The resolving power of the instrument was 8000.

UPLC conditions: a BEH C18 (100 mm × 2.1 mm, 1.7 μm) column (Waters) was used; the column temperature was maintained at 40 °C. Mobile phases A (water, 0.1% formic acid) and B (acetonitrile, 0.1% formic acid) were used; the gradient program was as follows: 0–4 min 5–25% B, 9–12 min 45–85% B, 14 min 100% B; 14.5–16 min 5% B, flow rate 0.35 mL min^−1^; injection volume 2 μL. The Acquity PDA detector wavelength was fixed at 210 nm, 254 nm, and 278 nm. The UPLC-Q/TOF MS column, gradient program, and PDA wavelengths were selected following the optimized conditions in a previous report [29].

A mixture of reference standards of MIAs and precursors (secologanin, ajmalicine, catharanthine, and vinblastine purchased from Sigma-Aldrich, St. Louis, MO, USA; vindoline and anhydrovinblastine purchased from Shanghai R&D Center for Standardization of Chinese Medicines, Shanghai, China) was detected and identified based on MS/MS spectra (Appendix A). Samples were applied in triplicate for quantification using calibration curves of the standards.

### 2.8. Soil pH Measurement

Soil samples from three plant pots of each Mg fertilizer treatment were thoroughly mixed to produce a single sample to ensure the reproducibility of results. The soil samples were air-dried at room temperature after removing the visible roots and organic residues, and then, they were naturally air-dried and passed through a 2 mm sieve for the measurement of the soil pH. Soil pH was assessed in a 1:2.5 soil: water (*w*/*v*) suspension using a pH meter (Fe28, FiveEasy Plus, MettlerToledo, Greifensee, Switzerland) and measured at 9 am. Three replicate readings were taken for each mixed sample.

### 2.9. Statistical Analysis

All experiments were conducted with three replicates. The trays for the seedling emergence experiment were arranged in a completely randomized design. By using SPSS (version 14.0, Chicago, IL, USA), the data were analyzed by one-way analysis of variance (ANOVA), followed by Duncan’s multiple range test for pairwise comparisons between groups and Pearson correlation analysis. *p*-values ≤ 0.05 were considered statistically significant. The Levene test was used for homogeneity of variance before ANOVA, and Dunnett’s test was also performed after ANOVA to identify significant differences among treatments (SI datasheet1).

## 3. Results

### 3.1. Seeds Germination

The application of MgO, MgSO_4_, and MgH_2_ promoted the seedling emergence. The seedling emergence under the MgO, MgSO_4,_ and MgH_2_ fertilization was 90.67%~91.82% while that of the control was 87.69%, but not statistically significant (Table 1). Time for 50% emergence was slightly affected by the application of MgO and MgH_2_ (Table 1).

### 3.2. Plant Growth

For plant height, the application of MgH_2_ and MgP exhibited a significant increase of 21% and 17% over the control, while MgO and MgSO_4_ fertilization didn’t show a significant difference from the control (Figure 1). MgH_2_ and MgP fertilization significantly increased plant weight by 60% and 29% over the control, respectively.

### 3.3. Chlorophyll Analysis

Compared to the control group, the MgP and MgH_2_-treated groups exhibited significant increases in total chlorophyll content by 13.5% and 12.7%, respectively (Figure 2A). In contrast, the MgSO_4_-treated group showed a marked reduction (19.0%). These differences were primarily driven by chlorophyll b content: MgP and MgH_2_ produced a significantly higher chlorophyll b content compared to CK, MgO, and MgSO_4_, whereas MgSO_4_ significantly decreased chlorophyll b relative to MgP and MgH_2_. Chlorophyll a content remained stable across Mg, MgO, and MgH_2_ treatments, but decreased significantly under MgSO_4_ treatment.

### 3.4. Magnesium Content

The Mg content in *C. roseus* leaves after fertilizer application is summarized in Figure 2B. The MgH_2_-treated group showed the highest Mg accumulation in leaves, with an increase of 48% over the control. Next was the MgP-treated group that elevated Mg content by 36%. Whereas MgO- and MgSO_4_-treated groups increased Mg content by 13% and 5%, respectively, though not statistically significant. On the whole, the MgH_2_- and MgP-treated groups had better effects than the MgO- and MgSO_4_-treated groups on improving Mg content in *C. roseus* leaves.

### 3.5. Soil pH

Soil pH was detected by potentiometry after the application of different Mg fertilizers. The results showed that the application of MgP, MgH_2_, and MgO led to a modest increase in soil pH (Figure 3). The soil pH of the control was 5.97. The pH remained relatively unchanged after the application of MgSO_4_, while the application of MgP and MgO raised the soil pH to 6.14~6.15. The soil pH in the MgH_2_ treatment group increased to 6.38, which was statistically significant.

### 3.6. Gene Expression in MIA Pathway

The expression levels of 18 key enzyme genes and transcription factor (TF) genes in the MIA biosynthetic pathway were measured under different treatments of Mg fertilizers by qPCR. Four fertilizers showed a positive effect on most gene expression, especially the expression of vindoline and vinblastine biosynthetic genes (*T16H*, *T3R*, *NMT*, *16OMT*, *D4H*, *DAT*, and *PRX1*) (Figure 4). MgH_2_ treatment significantly induced the expression of 14 genes and reduced the expression of 2 genes. MgP treatment led to a significant increase in the expression of 13 genes but a significant decrease in the expression of 4 genes. MgO treatment significantly upregulated 12 genes’ expression and downregulated 2 genes’ expression. The application of MgSO_4_ resulted in a significant increase in the expression of 10 genes, but significantly reduced the expression of 6 genes. Under the application of four fertilizers, 12 genes showed the highest expression in MgH_2_-treated group, including *TDC*, *GS*, *CS*, *TS*, *T16H*, *T3R*, *NMT*, *D4H*, *DAT*, *PRX1*, *CrEIN3* and *ORCA3* with maximum increase of 223%, 7%, 569%, 53%, 94%, 180%, 276%, 119%, 194%, 160%, 58% and 254% over the control, respectively. *Redox1* showed the highest expression in the MgP-treated group with an increase of 55%. *SGD* showed the highest expression in the MgO-treated group with an increase of 24%. *STR*, *PAS*, *16OMT* and *CrWRKY1* had the highest expression in MgSO_4_-treated group with increase of 39%, 34%, 250%, and 64%, while *SGD*, *GS*, *GO*, *Redox1*, *DPAS* and *TS* had the lowest expression with significant decrease of 38%, 26%, 50%, 22%, 60% and 48% over the control, respectively.

### 3.7. Alkaloids Analysis

After one month of fertilization, MgP, MgO, and MgH_2_ treatments showed a significant increase in the contents of catharanthine, vindoline, and ajmalicine compared to the control, while MgSO_4_ treatment only increased ajmalicine content but caused a significant decrease in the contents of catharanthine and vindoline (Figure 5). Among them, MgH_2_ treatment led to the highest contents of catharanthine, vindoline, and ajmalicine, increased by 67%, 49%, 417% over the control, respectively. MgP treatment increased the contents of catharanthine, vindoline, and ajmalicine by 33%, 34% and 192% over the control, respectively. MgO treatment also increased the contents of catharanthine, vindoline, and ajmalicine by 31%, 31% and 159% over the control, respectively. But in MgSO_4_ treatment, only the ajmalicine content increased by 89% over the control, and the catharanthine and vindoline contents decreased by 26% and 17%, respectively.

### 3.8. Correlation Analysis

Pearson correlation analysis showed that H_2_ release from fertilizers, soil pH, and Mg uptake of *C. roseus* had highly significant positive correlations (*p* < 0.001) with each other (Figure 6). Chlorophyll content, plant weight, and height of *C. roseus* exhibited significantly positive correlations (*p* < 0.01 or *p* < 0.05) with H_2_ release from fertilizers, soil pH, and Mg uptake. Seed emergence (SE) didn’t show a strong correlation with Mg content, H_2_ release, and soil pH. Time for 50% emergence (TFE) showed a significantly negative correlation with Mg content and H_2_ release. Mantel correlation analysis showed that the contents of vindoline, catharanthine, and ajmalicine exhibited highly significant positive correlations (*p* < 0.01) with Mg content, H_2_ release from fertilizers, soil pH, and chlorophyll content, but fewer correlations with SE, TFE, and plant weight (Figure 6).

Further correlation analysis showed that the genes expression of *TDC*, *CS*, *TS*, *T16H*, *T3R*, *NMT*, *D4H*, *DAT*, *ORCA3* and *CrEIN3* significantly positively correlated with H_2_ release, soil pH, Mg uptake and chlorophyll content, while the genes expression of *STR*, *SGD*, *Redox2*, *PAS*, *DPAS*, *OMT* and *CrMYC2* didn’t show significant correlations (Figure 7). Moreover, part of the MIA gene expression (*TDC*, *STR*, *DPAS*, *CS*, *TS*, *T16H*, *T3R*, *NMT*, *D4H*, *DAT*, *ORCA3*, *CrEIN3*, and *CrERF5*) showed significantly positive correlations with plant height (Figure 7). Overall, the correlation analysis revealed that MIA biosynthesis in *C. roseus* was significantly associated with H_2_ release, soil pH, Mg uptake, and chlorophyll content.

## 4. Discussion

As an essential element for plant growth, Mg fertilization is known to enhance crop yield and biomass [12,30]. Similarly, H_2_ has positive effects on seedling growth, adventitious rooting, root elongation, harvest freshness, stomatal closure, and anthocyanin synthesis [21]. The application of molecular hydrogen could increase the field and grain quality of rice [22]. In this study, application of MgH_2_, MgP, and MgO showed a significantly positive effect on the plant weight of *C. roseus* (Figure 1). Moreover, MgH_2_ and MgP fertilization positively affected the plant height of *C. roseus*. In contrast, MgSO_4_ had a slight effect on both plant weight and height of *C. roseus*. This aligns with previous reports that slow-release Mg fertilizers can improve crop yield more effectively than rapid-release alternatives [20]. Furthermore, *C. roseus* plants had a much better absorption of Mg nutrients under MgH_2_ and MgP application than under MgO and MgSO_4_ application (Figure 4). Pearson correlation analysis showed that Mg content of *C. roseus* leaves significantly positively correlated with H_2_ release. These results suggest a synergistic effect of Mg plus H_2_ fertilization. These fertilizers not only supply Mg^2+^ but also release H_2_ during hydrolysis, which ultimately led to the observed improvements in chlorophyll synthesis, plant biomass, and the upregulation of MIA biosynthetic genes, resulting in higher MIA accumulation. *C. roseus* Although seed emergence didn’t correlated with Mg uptake and H_2_ release, time for 50% emergence showed a significantly negative correlation with Mg uptake and H_2_ release. And H_2_ promoted cucumber seed germination by increasing sugar and starch metabolism [31]. This suggested that Mg plus H_2_ fertilization could promote seed germination earlier.

The role of Mg in photosynthesis is well-established as the central atom of chlorophyll [32,33]. Mg deficiency is interveinal chlorosis, meaning chlorophyll degradation due to excessive ROS production [34]. Consistently, we observed a strong positive correlation between Mg content and chlorophyll content in *C. roseus*. Mg plus H_2_ (MgP and MgH_2_) fertilization significantly increased both Mg and chlorophyll content, whereas Mg-only fertilization (MgO and MgSO_4_) had less effect on both in *C. roseus* (Figure 2). Both Mg and chlorophyll contents were positively correlated with H_2_ release. These results indicated that H_2_ release from Mg plus H_2_ fertilizers promoted Mg uptake, which in turn enhanced chlorophyll content in *C. roseus*. 

Moreover, H_2_ release was significantly positively correlated with soil pH (*p* < 0.01). MgH_2_ fertilization resulted in the highest soil pH and Mg content in *C. roseus* plants. Next is the MgP and MgO fertilization. The soil pH under the application of MgSO_4_ was as low as that of the control. Mg-slowly released fertilizers have a certain effect on improving soil acidity, which indirectly improves the utilization efficiency [20]. MgH_2_, MgP, and MgO all slowly produce Mg(OH)_2_ in soil. It has been reported that the stronger the acidity of the soil, the less conducive it is to maintaining the cations, so the soil replacement magnesium content is low [12]. MgSO_4,_ as a Mg-rapidly released fertilizer, is easier to leach Mg cation than Mg-slowly released fertilizers in soil [27]. Our result is consistent with the previous research. MgSO_4_ fertilization caused the lowest soil pH and Mg uptake among the four fertilizers.

In *C. roseus*, the iridoid pathway for the formation of secologanin and the indole pathway for the production of tryptamine are condensed to form 3α(S)-strictosidine under the catalysis of strictosidine synthase (STR) [1]. 3α(S)-strictosidine leads to the formation of catharanthine and tabersonine through a series of enzymatic reactions, including redox enzymes (Redox1), Tabersonine Synthase (TS), and α-hydrolase (CS) [35]. Tabersonine is converted into vindoline and catalysed by tabersonine 16-hydroxylase (T16H1/2), 16-hydroxytabersonine methyltransferase (16OMT), tabersonine 3-oxidase (T3O),tabersonine 3-O-acetyltransferase (T3R), N-methyltransferase (NMT), deacetoxyvindoline-4-hydroxylase (D4H), and deacetylvindoline 4-O-acetyltransferase (DAT) [36]. Finally, catharanthine and vindoline are condensed by peroxisomal thiolase to form vinblastine and vincristine [37]. Gene expression of the MIA biosynthesis is regulated by multiple environmental factors. It has been reported that transcript levels of *G10H*, *STR*, and *DAT* genes were up-regulated under potassium fertilization [5]. Light activates gene expression of the late stages of vindoline biosynthesis [38]. Our results revealed that the expression of most MIA biosynthetic genes, especially those in the vindoline pathway, exhibited positive correlations with Mg and chlorophyll contents in *C. roseus* leaves. As the central atom of the chlorophyll molecule in the light-absorbing complex of chloroplasts, Mg^2+^ may maintain a certain conformation of the antenna chromophore reaction centers and certain electron carriers on the molecular level and maintain their close contact at the molecular level to ensure efficient absorption, transfer, and conversion of light energy [12], which might facilitate the vindoline biosynthesis in *C. roseus*. Deficiencies in Mg were reported to reduce ajmalicine accumulation in *C. roseus* [11]. And Mg fertilizers have shown a positive effect on the alkaloid metabolism in tobacco, potato, *C. tangshen*, and other species [16,17,18,19]. Furthermore, H_2_ treatment upregulated key structural genes in the phenylpropanoid biosynthetic pathway and enhanced the accumulation of flavonoids and coumarins in the medicinal plant *Ficus hirta* [23].

Given the reported roles of Mg and H_2_ in influencing plant secondary metabolism, we explored whether their combination would yield synergistic effects on MIA biosynthesis. Our results suggested that Mg plus H_2_ fertilizers, MgH_2_ and MgP, had a better influence on promoting the MIAs accumulation than Mg-only fertilizers, MgO and MgSO_4_. MgH_2_ showed the strongest enhancement in the MIAs accumulation. Next is MgP. This change of alkaloid content was similar to gene expression. MgH_2_ could produce one more molecule of H_2_ than MgP. H_2_ has been reported to upregulate the expression of the JA receptor gene, OsCOI1, in rice [21,39]. Moreover, MIA contents had extremely significant positive correlations with H_2_ release, Mg content, and soil pH. All these results suggested that H_2_ released from MgH_2_ or MgP improved soil pH and Mg uptake, which further facilitated the growth and MIA biosynthesis in *C. roseus* (Figure 8). Combined with the slow release of Mg cation and H_2_, MgH_2_ can be used as an optimal fertilizer for *C. roseus* plant growth and MIA production.

## 5. Conclusions

In this study, Mg plus H_2_ fertilization (MgH_2_ and MgP) had a stronger effect than Mg alone fertilization (MgO and MgSO_4_) on promoting Mg uptake, plant growth, and MIA production in *C. roseus*. The results indicated that H_2_ released from MgH_2_ or MgP positively affected soil pH and Mg uptake, which further facilitated the growth and MIA biosynthesis in *C. roseus*. MgH_2_ fertilizer, due to its comprehensive effect on soil pH, Mg content, and plant growth, demonstrates great potential as an innovative fertilizer for *C. roseus* cultivation. These findings provide practical guidance for commercial production to enhance the yield of valuable anti-cancer alkaloids. Furthermore, this study lays the groundwork for future research to explore the application of Mg plus H_2_ fertilization in other medicinal plants under field conditions, *C. roseus.*

## Figures and Tables

**Figure 1 plants-14-03336-f001:**
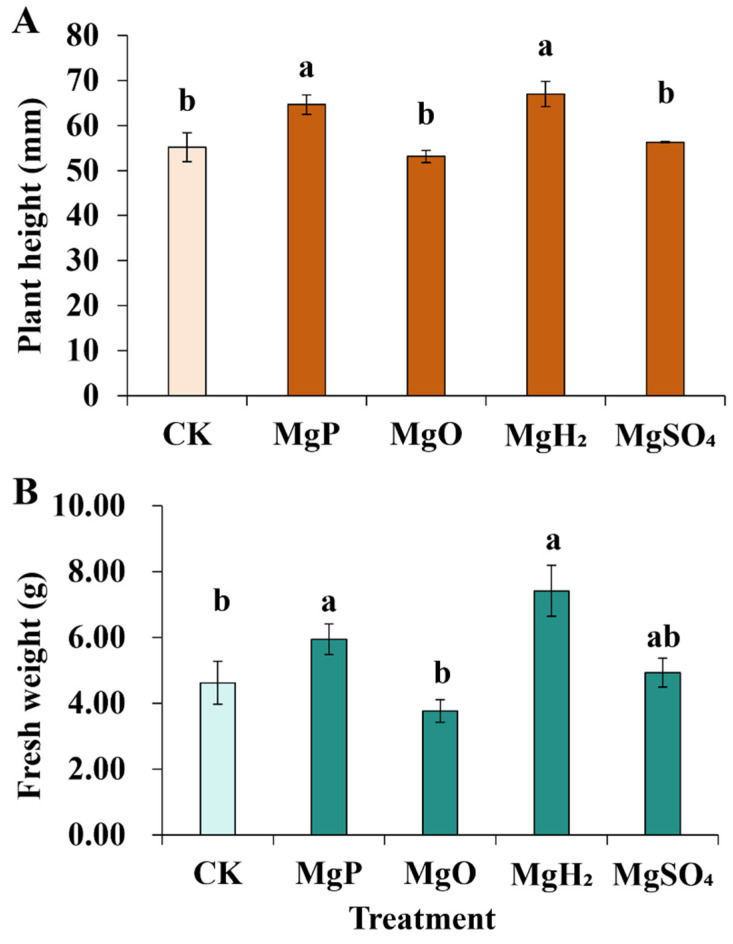
Plant height (**A**) and Fresh weight (**B**) of *C. roseus* after one month of Mg fertilizer application. Data are presented as mean ± standard error (SE). Statistical analysis was performed using one-way analysis of variance (ANOVA), followed by Duncan’s multiple range test for pairwise comparisons between groups. Different lowercase letters (a and b) in the figure indicate significant differences among different fertilizer treatment groups (*p* < 0.05). The sample size for each treatment group is n = 3 (i.e., 3 biological replicates were set for each treatment).

**Figure 2 plants-14-03336-f002:**
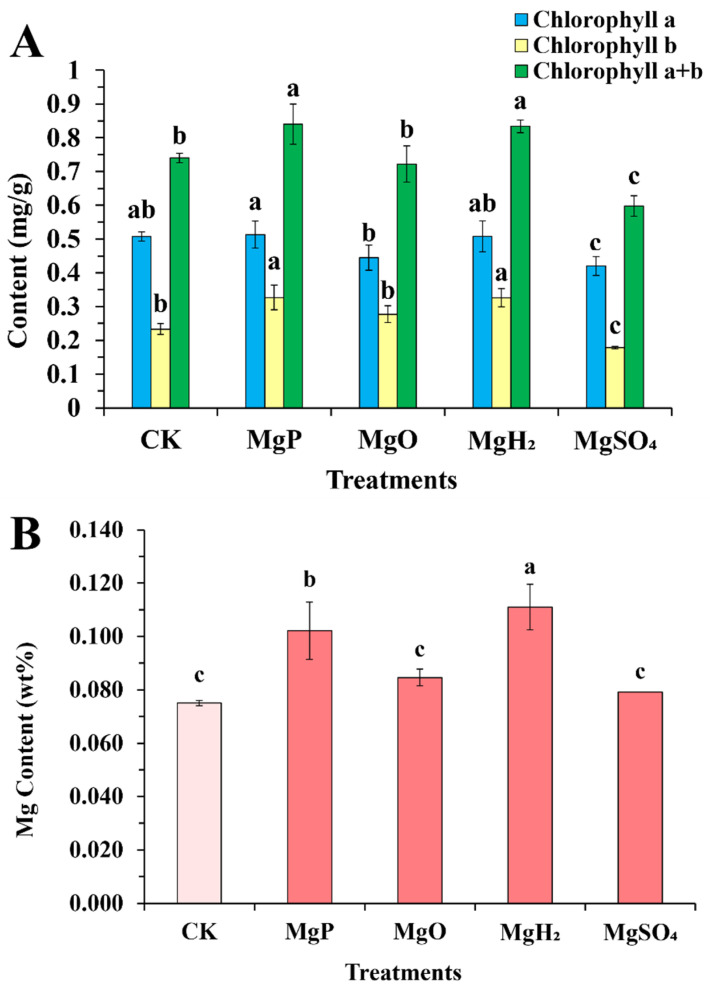
Chlorophyll content (**A**) and Mg content (**B**) in leaves of *C. roseus* after one month of four fertilizer applications. Data are presented as mean ± standard error (SE). Statistical analysis was performed using one-way analysis of variance (ANOVA), followed by Duncan’s multiple range test for pairwise comparisons between groups. Different lowercase letters (a–c) in the figure indicate significant differences among different fertilizer treatment groups (*p* < 0.05). The sample size for each treatment group is n = 3 (i.e., 3 biological replicates were set for each treatment).

**Figure 3 plants-14-03336-f003:**
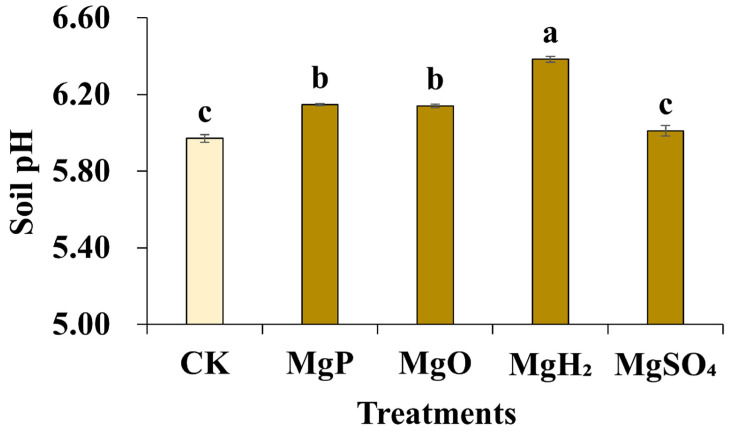
Soil pH after one month of Mg fertilizer application. Data are presented as mean ± standard error (SE). Statistical analysis was performed using one-way analysis of variance (ANOVA), followed by Duncan’s multiple range test for pairwise comparisons between groups. Different lowercase letters (a–c) in the figure indicate significant differences among different fertilizer treatment groups (*p* < 0.05). The sample size for each treatment group is n = 3 (i.e., 3 biological replicates were set for each treatment).

**Figure 4 plants-14-03336-f004:**
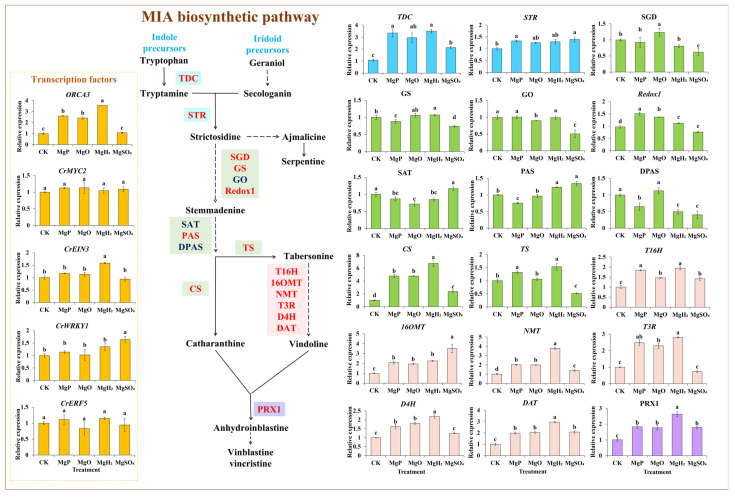
Effects of four Mg fertilizers on the MIA gene expression in *C. roseus*. Data are presented as mean ± standard error (SE). Statistical analysis was performed using one-way analysis of variance (ANOVA), followed by Duncan’s multiple range test for pairwise comparisons between groups. Different lowercase letters (a–e) in the figure indicate significant differences among different fertilizer treatment groups (*p* < 0.05). The sample size for each treatment group is n = 3 (i.e., 3 biological replicates were set for each treatment).

**Figure 5 plants-14-03336-f005:**
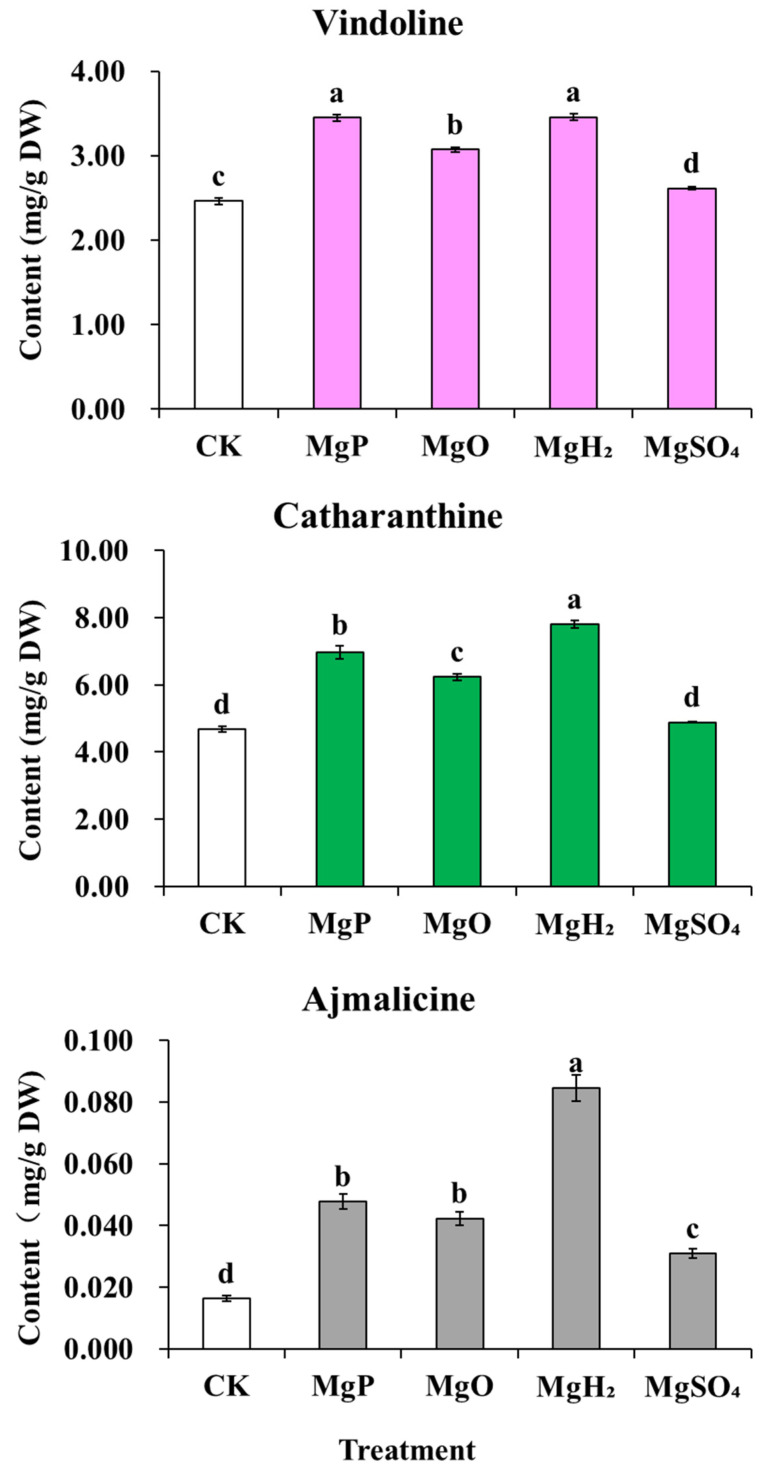
Effects of four Mg fertilizers on the MIA contents in *C. roseus*. Data are presented as mean ± standard error (SE). Statistical analysis was performed using one-way analysis of variance (ANOVA), followed by Duncan’s multiple range test for pairwise comparisons between groups. Different lowercase letters (a–d) in the figure indicate significant differences among different fertilizer treatment groups (*p* < 0.05). The sample size for each treatment group is n = 3 (i.e., 3 biological replicates were set for each treatment).

**Figure 6 plants-14-03336-f006:**
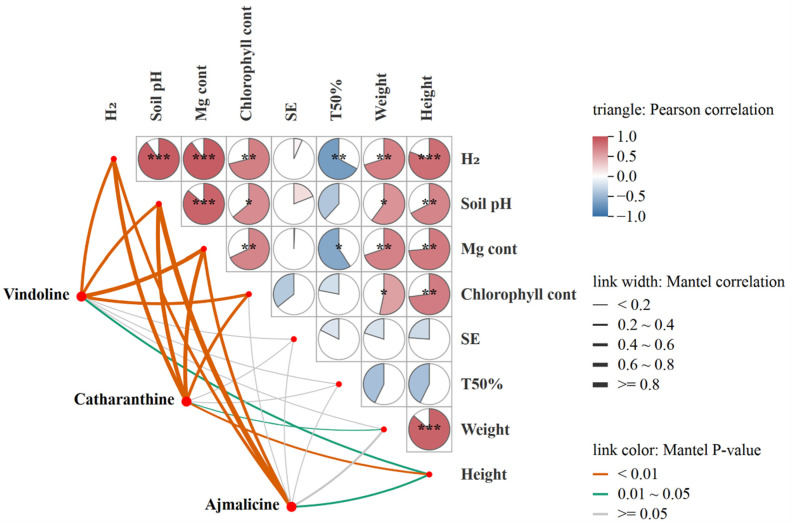
Pearson correlation and Mantel test between H_2_ release from fertilizers, soil pH, plant indexes, and the MIAs (vindoline, catharanthine, ajmalicine) contents. H_2_: Two molecules of H_2_ are released from one molecule of fertilizer. SE, seed emergence. TFE, Time for 50% emergence. Significance level, * *p* < 0.05, ** *p* < 0.01, *** *p* < 0.001.

**Figure 7 plants-14-03336-f007:**
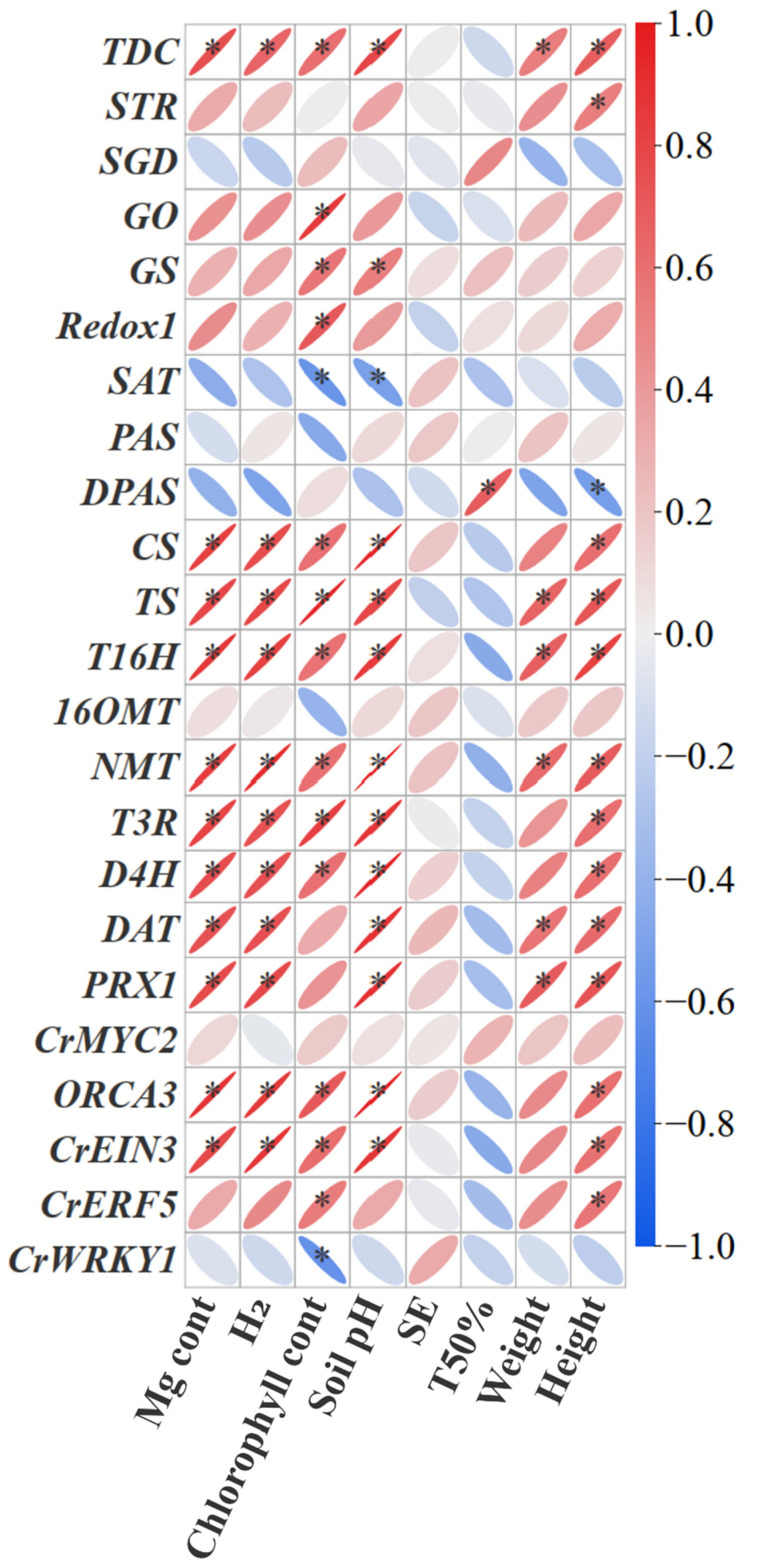
Pearson’s correlation analysis between plant indexes and MIA gene expression under Mg and H_2_ fertilization. The colour gradient and the degree of flattening of the ellipse indicate the correlation of the Pearson; red represents a positive correlation, and blue represents a negative correlation. H_2_: the molecule of H_2_ released from one molecule of fertilizer. SE, seed emergence. TFE, Time for 50% emergence. * indicates significance at *p* < 0.05.

**Figure 8 plants-14-03336-f008:**
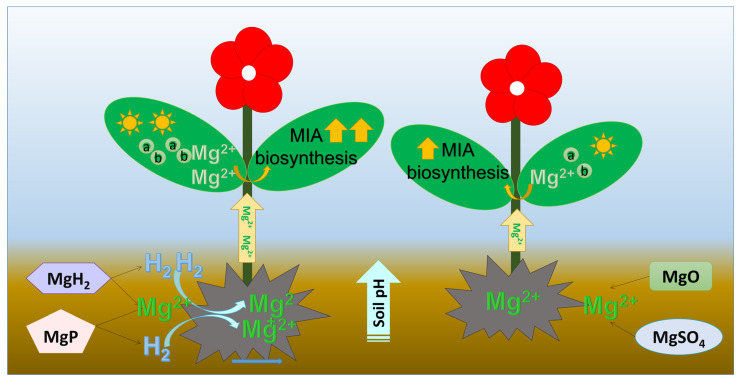
Scheme of the influence of Mg plus H_2_ fertilizers and Mg alone fertilizers on soil pH, Mg uptake, chlorophyll content, plant growth, and MIA biosynthesis. a: chlorophyll a; b: chlorophyll b.

**Table 1 plants-14-03336-t001:** Seedling emergence and Time for 50% emergence of *C. roseus* after four fertilizer applications.

Treatment	Seedling Emergence %	Time for 50% Emergence (Days)
CK	87.69 ± 0.27 ^a^	4.65 ± 0.07 ^ab^
MgP	88.01 ± 0.33 ^a^	4.50 ± 0.05 ^bc^
MgO	90.75 ± 0.27 ^a^	4.76 ± 0.12 ^a^
MgH_2_	90.67 ± 1.80 ^a^	4.46 ± 0.01 ^c^
MgSO_4_	91.82 ± 2.21 ^a^	4.58 ± 0.12 ^bc^

Data are presented as mean ± standard error (SE). Statistical analysis was performed using one-way analysis of variance (ANOVA), followed by Duncan’s multiple range test for pairwise comparisons between groups. Different lowercase letters (a–c) in the figure indicate significant differences among different fertilizer treatment groups (*p* < 0.05). The sample size for each treatment group is n = 3 (i.e., 3 biological replicates were set for each treatment).

## Data Availability

All the data needed to evaluate the conclusions in the paper are present in the paper and/or the Appendix A.

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
