# Peer review of "Magnesium Plus Hydrogen Fertilization Enhances Mg Uptake, Growth Performance and Monoterpenoid Indole Alkaloid Biosynthesis in Catharanthus roseus"

_plants, 2025, doi:10.3390/plants14213336_

Round 1
Reviewer 1 Report
Comments and Suggestions for Authors
In this manuscript, the enhancements of magnesium plus hydrogen fertilization on Mg uptake and growth performance were confirmed. Furthermore, the biosynthesis of the monoterpenoid indole alkaloid was increased. These findings were important.
However, there were some issues to be addressed before publication.
- As shown in Table 1, the seedling emergence of the control CK group was the lowest. However, the lowest value was observed for the MgO group as shown in Fig. 1A. These data in the figure and table were inconsistent. Additionally, it is better to give a full name of CK in the preceding paragraph of Table 1.
- As shown in Fig. 2B, the Mg content in leaves of the MgSO4 group was almost the same as that of the control CK group. As Discussion reported ‘Mg content had a significantly positive correlation with chlorophyll content’, the chlorophyll content of the MgSO4 should be almost the same that of the control CK group. However, Fig. 2A showed the MgSO4 group had a sharp decline of chlorophyll content, compared with that of the control group.
- The title of Fig. S2 should be revised. As shown in Fig. S2, the calibration curves of tryptamine, secologanin, anhydrovinblastine and vinblastine were not provided.
Author Response
Reply to reviewer 1
Reviewer1
In this manuscript, the enhancements of magnesium plus hydrogen fertilization on Mg uptake and growth performance were confirmed. Furthermore, the biosynthesis of the monoterpenoid indole alkaloid was increased. These findings were important.
However, there were some issues to be addressed before publication.
- As shown in Table 1, the seedling emergence of the control CK group was the lowest. However, the lowest value was observed for the MgO group as shown in Fig. 1A. These data in the figure and table were inconsistent. Additionally, it is better to give a full name of CK in the preceding paragraph of Table 1.
Response 1:
Thank you for your comments. We believe there might be a misunderstanding. Figure 1A displays plant height, where the MgO group showed the lowest value, while Table 1 summarizes seedling emergence rate, where the control (CK) group was the lowest. There is no inconsistency between the figure and the table, as they measure different aspects of plant growth. We have modified as “The seedling emergence under the MgO, MgSO4 and MgH2 fertilization were 90.67% ~ 91.82% while that of the control was 87.69%, but not statistically significant (Table 1). Time for 50% emergence significantly reduced only under the application of MgH2 (Table 1).”
- As shown in Fig. 2B, the Mg content in leaves of the MgSO4 group was almost the same as that of the control CK group. As Discussion reported ‘Mg content had a significantly positive correlation with chlorophyll content’, the chlorophyll content of the MgSO4 should be almost the same that of the control CK group. However, Fig. 2A showed the MgSO4 group had a sharp decline of chlorophyll content, compared with that of the control group.
Response 2:
Thank you for your comments. We used the data of Mg content and chlorophyll content from all the treatments (MgSO4, MgO, MgH₂, MgP and the control) to perform Pearson correlation analysis by SPSS software. Although the MgSO₄ group exhibited a sharp decline in chlorophyll content that did not align with its Mg content, this single inconsistency did not affect the overall correlation analysis.
- The title of Fig. S2 should be revised. As shown in Fig. S2, the calibration curves of tryptamine, secologanin, anhydrovinblastine and vinblastine were not provided.
Response 3:
Thank you for your comments. As recommended, we have revised the title of Fig. S2.

Reviewer 2 Report
Comments and Suggestions for Authors
Overall Feedback
The manuscript presents a novel study on the combined effects of magnesium (Mg) and hydrogen (H₂) fertilization on Catharanthus roseus, examining growth, Mg uptake, chlorophyll content, gene expression, and alkaloid production. The work addresses an important and underexplored topic in medicinal plant research, and the results are generally robust, detailed, and well-supported by informative figures and tables. The study has clear practical relevance, particularly for strategies aimed at enhancing secondary metabolite production.
However, several key aspects require improvement. The abstract should include quantitative results and clearly state the research gap, while the introduction would benefit from a stronger logical flow, explicit emphasis on novelty, and a clearly stated hypothesis. Materials and Methods need clarification regarding Mg dose standardization, replication, PCR conditions, and statistical assumptions to ensure reproducibility. In the Results, consistency in treatment naming, precise statistical language, an ANOVA summary table, and clarified figure legends are needed. The Discussion could condense repetitive mechanistic explanations, better integrate literature on H₂ effects, and improve transitions between treatments. Finally, the Conclusions should be concise, emphasize practical significance, and include limitations or future research directions.
Overall, the manuscript is scientifically sound and of potential high impact. While the study is generally robust and well-conducted, both major and minor issues need to be addressed to improve clarity, methodological transparency, and overall presentation.
Recommendation: Major Revision.
Below, detailed comments and suggestions are provided for each section of the manuscript.
Comments on the Abstract
- Opening sentence
- The statement “Magnesium and hydrogen fertilization is crucial…” is overly general and not specific to C. roseus. It is recommended to revise it to a more precise formulation that reflects known trends without overgeneralizing, e.g., “Magnesium and hydrogen fertilization have been reported to contribute to plant growth and metabolite production.”
- Second sentence
- The phrase “However, simultaneous fertilization … has less explored” contains a grammatical error (“has been less explored”) and does not clearly identify the research gap. The authors should explicitly explain why investigating this combination in C. roseus is important, highlighting the relevance of H₂ release for secondary metabolite (alkaloid) production.
- Experimental design (description of treatments)
- The abstract mentions five treatments (MgO, MgSO₄, MgH₂, magnesium powder (MgP), and control), which is appropriate. To improve clarity, a brief phrase indicating that these treatments represent different mechanisms of Mg and H₂ release (e.g., slow- vs. rapid-release sources) could be added, without going into detailed rationale.
- Results
- Key quantitative data are missing. At least one or two concrete numerical results (e.g., percentage increase in biomass or alkaloid content) should be included. The expression “greater increases” is vague and should be replaced with precise outcomes.
- The reported change in soil pH (“raised soil pH from 6.00 to 6.14”) is minimal; it is unclear whether this change is statistically or biologically significant. The authors should indicate whether the difference is significant and relevant to plant performance.
- Conceptual clarity
- The statement “Mg uptake, H₂ release and soil pH were positively correlated…” is a valid result. In the abstract, it could be phrased more concisely to highlight the significant statistical correlations without appearing as discussion. For example: ‘Pearson correlation analysis showed significant positive correlations among H₂ release, soil pH, and Mg uptake, as well as with plant growth and MIA content.
- Conclusion
- The phrase “These findings suggest…” is acceptable, but the sentence is long and somewhat repetitive. The conclusion could be strengthened and made more concise by emphasizing the practical significance. For example: “highlighting the potential of MgH₂ and Mg powder as innovative fertilizers to enhance alkaloid yields in medicinal plants.” This keeps the key message clear and impactful.
Overall Summary – Abstract
- The abstract should clearly articulate the research gap, include at least one or two key quantitative results (e.g., percentage increase in biomass or alkaloid content), and simplify the conclusion for stronger impact.
- Avoid vague expressions (e.g., “greater increases”) and focus on concise reporting of main findings rather than interpretations.
- Simplify and streamline the conclusion to highlight practical significance.
Comments on the Introduction
- Pharmacological background (lines 39–48)
- While informative, several expressions are not in proper academic English:
- “have been wellknown” → “are well known”
- “The small amounts… makes semi-synthetic route…” → requires subject–verb agreement (“make”).
- The introduction does mention the need to increase MIA yield and accumulation, linking pharmacological importance to cultivation. However, this connection could be made more explicit and smoother, emphasizing that optimizing cultivation strategies is critical to increase the availability of pharmacologically valuable MIAs.
- Literature review on fertilization (lines 49–64)
- The review reads as a fragmented list. Stronger logical connections are needed, for example: “Taken together, these studies indicate that nutrient supply can modulate MIA biosynthesis, although often at the expense of growth performance.”
- A concluding paragraph should emphasize the research gap: despite extensive studies on N, K, Ca, and Zn, few have investigated Mg, and none have explored the combined role of Mg and H₂, highlighting the novelty of the present study.
- Section on magnesium (lines 65–76)
- Provides useful background but lacks a clear transition to C. roseus. References to potato and tobacco are informative but should be linked to the present study, for example: “These findings suggest that Mg fertilization can influence alkaloid metabolism in various plant species, yet its role in MIA production in C. roseus remains unclear.”
- If available, any studies on Mg effects specifically in C. roseus should be mentioned.
- Section on hydrogen (lines 77–89)
- The discussion on hydrogen is informative but overly technical; details on reaction kinetics (“extremely slow reaction kinetics…”) could be condensed or relocated to the Methods or Discussion section.
- More importantly, the introduction of hydrogen biology in plants is currently limited to seed germination and hormone regulation. For the present study, which aims to examine the effects of Mg + H₂ fertilization on MIA production in C. roseus, it is critical to include references to prior studies investigating the effects of hydrogen on secondary metabolites and/or on genes and enzymes involved in their biosynthetic pathways.
- Furthermore, if available, the authors should cite prior literature on the combined effects of Mg + H₂ on plant growth, development, and secondary metabolite accumulation in medicinal plants. This would provide a stronger rationale for the study and better situate it within the existing research context.
- If no such studies exist, this gap further highlights the novelty and relevance of the present investigation.
- Final paragraph / study objective (lines 90–93)
- The closing paragraph appropriately describes the study objectives, but it lacks a clearly stated hypothesis. Formulating an explicit hypothesis would improve clarity. For example: “We hypothesized that MgH₂ and Mg powder, by simultaneously supplying Mg and releasing H₂, would improve Mg uptake, plant growth, and MIA biosynthesis in C. roseus compared to conventional Mg fertilizers.”
- Additionally, adding a transition sentence linking the research gap to the study objective would strengthen the rationale. For example: “Given the limited information on hydrogen’s effects on secondary metabolites and the lack of studies on combined Mg + H₂ application in medicinal plants, it is important to evaluate their impact on C. roseus growth and MIA production.”
- These additions would make the final paragraph more explicit, logical, and readable, clearly connecting background, gap, and study aim.
Overall Summary – Introduction
- Improve logical flow and connections between previous studies.
- Emphasize the research gap in Mg and Mg+H₂ fertilization.
- Reduce overly technical chemical details; include references on hydrogen’s effects on secondary metabolism.
- Formulate a clear working hypothesis and provide a smooth transition from the identified research gap to the study aim.
Comments on Materials and Methods
- Section 2.1 – Plant Materials and Treatments
- The description of the plant material and fertilizer treatments is generally clear. However, the rationale for selecting the four Mg fertilizers could be briefly stated in the Methods. In the Introduction, MgO and MgSO₄ were described as conventional slow- and rapid-release Mg sources, while MgH₂ and Mg powder were introduced as novel Mg + H₂ fertilizers. Explicitly restating this classification in Section 2.1 would improve coherence and help readers understand the comparative design of the study.
- The concentration of 7 mg/kg soil is provided, but the method for calculating or standardizing this dose across different fertilizer forms is not fully explained. Including a brief note on how equivalent Mg concentrations were achieved would improve reproducibility.
- While it is specified that 20 plantlets per treatment were collected for subsequent analyses, the number of pots or experimental units from which these plants were obtained is not indicated. Clarifying this point would strengthen transparency regarding biological replication and the independence of samples.
- Section 2.2 – Seedling Emergence
- The methodology is detailed, but the explanation of the t50% equation could be formatted more clearly. Some variables (e.g., ed - ed-1) are confusing as currently written.
- It would be useful to specify how and when emergence counts were performed (e.g., time of day) and whether observers were blinded to treatments to reduce potential bias.
- “The trays were arranged in a completely randomized design” this information would be more appropriate in the Statistical Analysis section.
- Section 2.3 – Plant Height and Weight
- Methods are generally clear, but it is recommended to specify whether measurements were taken on fresh or dry weight (or both) for clarity and consistency. For example, in the Results, plant weight is reported as fresh weight, whereas chlorophyll, Mg, and alkaloid concentrations are measured on powdered leaf samples, with alkaloids expressed per dry weight. Clarifying the basis of all measurements would help readers correctly interpret and compare the data.
- The number of plantlets per replicate is given (five), but indicate whether these five were pooled for weight or measured individually, as this affects statistical analysis.
- Section 2.4 – Chlorophyll Analysis
- The extraction and measurement procedure is adequately described. However, “grinded” should be corrected to ground.
- The protocol mentions ELISA reader for absorbance measurement; typically, chlorophyll is measured by spectrophotometer. Clarify whether the ELISA reader was used as a plate reader for absorbance, and provide model specifics if possible.
- Methods report readings at 646 nm but the Arnon equations are written with A₆₄₅. Please state which absorbance was actually used for calculations (A₆₄₅ or A₆₄₆) and justify the interchange if applicable.
- Section 2.5 – Mg Content Determination
- Overall method is clear. Recommend specifying the digestion vessel type (glass/teflon) and whether blanks and standards were included for ICP-OES calibration.
- Clarify the sample preparation step: “moistened with a small amount of water” is vague; provide exact volume or procedure for reproducibility.
- Section 2.6 – qRT-PCR
- Procedure is generally clear, but the storage temperature is given as -20 °C; for RNA stability, -80 °C is more typical. Confirm accuracy.
- Although primers are listed in Supplementary Table S1, indicate the amount of RNA used for cDNA synthesis to ensure reproducibility.
- Include details about PCR cycling conditions (temperatures, times, number of cycles) to enhance reproducibility.
- Section 2.7 – MIAs Analysis
- The UPLC-Q/TOF-MS methodology is comprehensive and detailed.
- It would be helpful to briefly explain the rationale for the choice of column, gradient program, and PDA detector wavelengths, to clarify why these conditions were optimal for MIAs separation and quantification.
- Section 2.8 – Soil pH Measurement
- Methods are generally adequate, but clarify at which time point(s) soil pH was measured and whether multiple readings were taken per mixed sample to ensure reproducibility.
- Section 2.9 – Statistical Analysis
- Use of ANOVA and Pearson correlation is appropriate.
- Recommend clarifying whether data were tested for normality and homogeneity of variance before ANOVA (specify the type of test used: graphical tests and/or analytical).
- Specify whether multiple comparison tests (e.g., Tukey’s HSD, Sidak, Fisher LSD etc) were performed after ANOVA to identify significant differences among treatments.
Overall Summary - Materials and Methods
- Methods are generally well described, but some sections could benefit from clarification for reproducibility: e.g., Mg dose standardization, sample preparation details, RNA quantity, PCR cycling, and replication strategy.
- Minor corrections needed for language (e.g., “grinded” → ground) and consistency (wavelengths, notation).
- Consider adding more details on quality controls, standard curves, and statistical assumptions to strengthen rigor and reproducibility.
Comments on Results
- Section 3.1 – Seed Germination
- The results are clearly presented, including both percentage emergence and T50% values.
- Minor suggestion: the text says improvements were “not significantly” – consider rephrasing as “not statistically significant” for clarity in academic writing.
- Section 3.2 – Plant Growth
- The results are clearly presented, and reporting percentage increases is informative.
- The distinction in effects between MgH₂/MgP and MgO/MgSO₄ treatments is clear and well supported by the data.
- Section 3.3 – Chlorophyll Analysis
- The trends for chlorophyll a and b are described well, and percentage changes are provided.
- Minor language corrections: “grinded” → ground (if repeated here from Methods).
- Suggest clarifying the Results text by replacing ‘Mg’ with ‘MgP’ for consistency with the treatment names used in the Methods (MgSO₄, MgO, MgH₂, MgP), to avoid ambiguity regarding which magnesium treatment caused the observed effects on chlorophyll b.
- Suggest using precise statistical language when reporting comparisons. For example, replace ‘Mg and MgH2 treatments significantly enhanced chlorophyll b, while MgSO4 caused a sharp decline’ with ‘MgP and MgH2 produced a significantly higher chlorophyll b content compared to CK, MgO, and MgSO4, whereas MgSO4 significantly decreased chlorophyll b relative to MgP and MgH2.’ This statistically accurate language should be applied consistently throughout the entire Results section to clearly indicate which differences are significant.
- Section 3.4 – Magnesium Content
- Mg accumulation is clearly reported with percentage increases; the ranking of fertilizers is evident.
- Minor suggestion: replace “but both not significantly” with “though not statistically significant” for correct academic phrasing.
- Section 3.5 – Soil pH
- Results are adequately reported, with numerical pH values.
- Suggest clarifying the phrasing: “a certain increase” → “a modest increase” or “an increase of X units”.
- Consider explicitly stating whether the MgH₂-induced increase (to 6.38) was statistically significant compared to the control.
- Fig. 3: Maintain consistency in naming treatments for all figures (Mg or MgP); same in the text
- Section 3.6 – Gene Expression in MIA Pathway
- Very detailed reporting of gene expression changes. Percent increases over control are provided, which is excellent.
- Suggest minor rewording for readability: long lists of percentages could be summarized as “Most genes in the MIA pathway were significantly upregulated under MgH2 and MgP treatments, with maximum increases of 569% (Redox1) and 223% (TDC), respectively (Fig. 4).” Detailed values can remain in the figure legend or supplementary table.
- Consider consistency in gene naming (e.g., 16OMT vs T16OMT) and confirm all genes mentioned match Table S1.
- Section 3.7 – Alkaloids Analysis
- The results are clearly presented, with fold changes and percentage increases that highlight treatment effects.
- Suggest minor stylistic adjustments: instead of “And MgO treatment increased…”, use “MgO treatment also increased…”
- The comparison of MgSO4 with other treatments is clear and highlights the contrasting effect; well presented.
- Additional suggestion: for consistency across the Results section, consider reporting changes either uniformly as percentages or as fold changes, rather than mixing both formats. This would improve clarity and readability for the audience.
- Section 3.8 – Correlation Analysis
- Correlation analyses are well described. Positive and negative correlations are clearly reported with significance levels.
- Suggest clarifying some phrasing: “extremely significantly positively correlated” → “highly significant positive correlations (p < 0.001)” for standard academic style.
- Consider grouping correlation results by category for clarity (e.g., growth traits, chlorophyll/Mg, gene expression, MIAs).
- The link between gene expression, plant traits, and H2 release is well presented; possibly add a brief summary sentence highlighting the main findings from Figures 6–7.
- Additional suggestion – ANOVA results table
it would be highly beneficial for transparency and reproducibility if the authors provide a summary table of the ANOVA results for all measured variables.
The table could include:
- F-values for each variable,
- Corresponding p-values,
- Symbols (e.g., *, **, ***) for different significance levels, and
- “ns” for non-significant comparisons.
This table could be included either in the main text (if concise) or in the Supplementary Materials to avoid overloading the Results section. Providing such a table would enhance clarity, allow readers to quickly assess the statistical robustness, and strengthen the overall presentation of the study.
- Additional suggestion – Figure legends
- It is recommended that the authors specify in each figure legend the type of multiple comparison test used (e.g., Tukey’s HSD, LSD) and the sample size for each treatment (n = …). Including this information would improve clarity and reproducibility, allowing readers to fully understand the statistical analyses underlying the reported significant differences.
- Additional suggestion – Consistency in style and significance reporting
- Ensure consistency in the notation of chemical formulas (e.g., MgH₂ vs MgH2) and abbreviations (SE, TFE, MIAs).
Overall Summary – Results (revised)
- Results are robust, detailed, and generally well-structured.
- Ensure consistent naming of treatments (MgP, MgO, MgH₂, MgSO₄) throughout text, figures, and tables.
- Use precise statistical language when reporting comparisons, indicating which differences are significant.
- Provide an ANOVA summary table for transparency.
- Clarify figure legends: specify sample size (n) and multiple comparison tests used.
- Rephrase ambiguous or informal phrases (e.g., “grinded” → “ground”, “a certain increase” → “modest increase of X units”).
- Summarize extensive gene expression and alkaloid results for readability, while keeping detailed values in figure legends or supplementary tables.
- Ensure consistency in units, abbreviations, and notation across the Results section.
- Maintain this precise and statistically accurate language consistently throughout the Results.
Comments on Discussion
- General structure and clarity:
The discussion remains generally well-organized and logically links the results to previous studies on Mg and H₂ fertilization. Some sentences are still long and could be split for clarity. Statements largely distinguish between results and interpretations now, but minor adjustments could further separate reporting from mechanistic explanation. Consider also maintaining consistent terminology (e.g., MgH₂ vs MgH2, MIA vs MIAs). - Integration with literature:
References on Mg effects are adequate, and H₂ literature is cited. However, references on hydrogen’s role specifically in secondary metabolite biosynthesis or regulation of biosynthetic genes in plants (especially in roseus or medicinal plants) are still sparse. Adding these would strengthen the rationale and highlight the novelty of the Mg + H₂ combination. - Mechanistic explanations:
The MIA biosynthetic pathway discussion is thorough. Repetitive details on Mg’s role in chlorophyll/light absorption can be condensed or integrated more smoothly with gene expression and alkaloid results to improve readability. - Statistical and quantitative clarity:
Quantitative data and statistical significance have already been appropriately presented in the Results. In the Discussion, ensure statements such as “positive effect” or “better absorption” are explicitly linked to the relevant figures/tables rather than repeating numbers unnecessarily. - Figures and interpretation:
Figure 8 is now properly referenced in the text when interpreting results, so this point has been addressed. - Transitions and flow:
Transitions could still be slightly improved. For instance, after discussing Mg effects, a short sentence highlighting the rationale for adding H₂ would make the link to Mg + H₂ effects more explicit. - Language and style:
Minor language edits are suggested: e.g., “had slightly effect” → “had a slight effect”; “similar as gene expression” → “similar to gene expression.” Avoid repetitive phrases like “significantly positive correlation” repeated multiple times.
Overall Summary – Discussion
- Well-organized, logically links results to literature.
- Include references on H₂ effects on secondary metabolism and gene regulation in medicinal plants, ideally including C. roseus.
- Strengthen transitions between discussion of Mg alone, H₂ alone, and Mg + H₂ combination.
- Condense repetitive mechanistic explanations to improve readability.
- Ensure all qualitative statements are clearly linked to figures/tables.
- Minor language adjustments recommended.
Comments on Conclusions
Content clarity:
The conclusions accurately summarize the main findings: Mg + H₂ fertilizers (MgH₂ and MgP) outperform Mg-alone fertilizers in promoting Mg uptake, plant growth, and MIA production.
Practical implications:
The potential application for commercial cultivation is clearly stated, reinforcing applied relevance.
Language and phrasing:
Minor improvements: e.g., “These findings have practical guidance to commercial production” → “These findings provide practical guidance for commercial production.” Sentences could be made slightly more concise and avoid repeating mechanistic details.
Future directions and limitations: Conclusions lack future research directions or limitations. I recomende to include them.
Overall Summary – Conclusions
- Concisely summarize findings.
- Emphasize novelty and practical impact of Mg + H₂ fertilization.
- Minor language improvements for clarity.
- Add a brief note on future research directions or limitations to frame the study within a broader context.
Author Response
Response to reviewer 2
Reviewer2:
Overall Feedback
The manuscript presents a novel study on the combined effects of magnesium (Mg) and hydrogen (H₂) fertilization on Catharanthus roseus, examining growth, Mg uptake, chlorophyll content, gene expression, and alkaloid production. The work addresses an important and underexplored topic in medicinal plant research, and the results are generally robust, detailed, and well-supported by informative figures and tables. The study has clear practical relevance, particularly for strategies aimed at enhancing secondary metabolite production.
However, several key aspects require improvement. The abstract should include quantitative results and clearly state the research gap, while the introduction would benefit from a stronger logical flow, explicit emphasis on novelty, and a clearly stated hypothesis. Materials and Methods need clarification regarding Mg dose standardization, replication, PCR conditions, and statistical assumptions to ensure reproducibility. In the Results, consistency in treatment naming, precise statistical language, an ANOVA summary table, and clarified figure legends are needed. The Discussion could condense repetitive mechanistic explanations, better integrate literature on H₂ effects, and improve transitions between treatments. Finally, the Conclusions should be concise, emphasize practical significance, and include limitations or future research directions.
Overall, the manuscript is scientifically sound and of potential high impact. While the study is generally robust and well-conducted, both major and minor issues need to be addressed to improve clarity, methodological transparency, and overall presentation.
Response:
Thank you for your comments. We have revised in details as follows.
Recommendation: Major Revision.
Below, detailed comments and suggestions are provided for each section of the manuscript.
Comments on the Abstract
- Opening sentence
- The statement “Magnesium and hydrogen fertilization is crucial…” is overly general and not specific to C. roseus. It is recommended to revise it to a more precise formulation that reflects known trends without overgeneralizing, e.g., “Magnesium and hydrogen fertilization have been reported to contribute to plant growth and metabolite production.”
Response 1:
Thank you for your comments. We have revised the sentence as suggested, and replaced the original sentence with “Magnesium and hydrogen fertilization have been reported to contribute to plant growth and metabolite production.”
- Second sentence
- The phrase “However, simultaneous fertilization … has less explored” contains a grammatical error (“has been less explored”) and does not clearly identify the research gap. The authors should explicitly explain why investigating this combination in C. roseus is important, highlighting the relevance of H₂ release for secondary metabolite (alkaloid) production.
Response 2:
Thank you for your comments. We have revised the sentence as suggested, and replaced the original sentence with “Simultaneous fertilization of magnesium and hydrogen is a promising strategy for plant development and secondary metabolism but remains unexplored in Catharanthus roseus, ……”
- Experimental design (description of treatments)
- The abstract mentions five treatments (MgO, MgSO₄, MgH₂, magnesium powder (MgP), and control), which is appropriate. To improve clarity, a brief phrase indicating that these treatments represent different mechanisms of Mg and H₂ release (e.g., slow- vs. rapid-release sources) could be added, without going into detailed rationale.
Response 3:
Thank you for your comments. Considered the word limitation of the abstract, we have added the necessary information in the section of both “Introduction” and “Materials and methods” to improve clarity.
- Results
- Key quantitative data are missing. At least one or two concrete numerical results (e.g., percentage increase in biomass or alkaloid content) should be included. The expression “greater increases” is vague and should be replaced with precise outcomes.
- The reported change in soil pH (“raised soil pH from 6.00 to 6.14”) is minimal; it is unclear whether this change is statistically or biologically significant. The authors should indicate whether the difference is significant and relevant to plant performance.
Response 4:
Thank you for your comments. We have added one concrete numerical result of alkaloid content as “Among them, MgH₂ yielded the highest contents of catharanthine, vindoline and ajmalicine, reaching 1.67-, 1.49- and 5.17-fold of the control, respectively.” And we have removed the vague expression “greater increases”. Instead, we used “MgH₂ and MgP significantly elevated Mg content, chlorophyll content, plant height and weight over MgO, MgSO₄ and the control.” To indicate that the difference of soil pH is significant and relevant to plant performance, we have rewritten the sentence as “Application of MgH₂, MgP and MgO fertilizers significantly raised soil pH to 6.14~6.38.”
- Conceptual clarity
- The statement “Mg uptake, H₂ release and soil pH were positively correlated…” is a valid result. In the abstract, it could be phrased more concisely to highlight the significant statistical correlations without appearing as discussion. For example: ‘Pearson correlation analysis showed significant positive correlations among H₂ release, soil pH, and Mg uptake, as well as with plant growth and MIA content.
Response 5:
Thank you for your comments. We have replaced the original sentence with “Pearson correlation analysis showed significant positive correlations among H₂ release, soil pH, and Mg uptake, as well as with plant growth and MIA content.”
- Conclusion
- The phrase “These findings suggest…” is acceptable, but the sentence is long and somewhat repetitive. The conclusion could be strengthened and made more concise by emphasizing the practical significance. For example: “highlighting the potential of MgH₂ and Mg powder as innovative fertilizers to enhance alkaloid yields in medicinal plants.” This keeps the key message clear and impactful.
Response 6:
Thank you for your comments. We have strengthened the sentence and made it more concise as suggested, and replaced the original sentence with “These findings suggest that Mg plus H2 fertilizers released H2 and increased soil pH to promote Mg uptake, chlorophyll contents, plant growth and MIA biosynthesis in C. roseus, highlighting the potential of MgH₂ and Mg powder as innovative fertilizers to enhance alkaloid yields in medicinal plants.”
Overall Summary – Abstract
- The abstract should clearly articulate the research gap, include at least one or two key quantitative results (e.g., percentage increase in biomass or alkaloid content), and simplify the conclusion for stronger impact.
- Avoid vague expressions (e.g., “greater increases”) and focus on concise reporting of main findings rather than interpretations.
- Simplify and streamline the conclusion to highlight practical significance.
Response:
Thank you for your comments. Following the comments, we have revised the abstract, clearly articulated the research gap, included the key quantitative result of alkaloid content, removed the vague expression, and simplified the conclusion to highlight practical significance.
Comments on the Introduction
- Pharmacological background (lines 39–48)
- While informative, several expressions are not in proper academic English:
- “have been wellknown” → “are well known”
- “The small amounts… makes semi-synthetic route…” → requires subject–verb agreement (“make”).
- The introduction does mention the need to increase MIA yield and accumulation, linking pharmacological importance to cultivation. However, this connection could be made more explicit and smoother, emphasizing that optimizing cultivation strategies is critical to increase the availability of pharmacologically valuable MIAs.
Response 1:
Thank you for your comments, We have revised the sentence as suggested, and replaced the original sentence with “are well known” and “The small amounts… make semi-synthetic route…”
Thank you for your comments, We have revised the sentence as suggested, We added “thus”, “Therefore” to strengthen the causality and logicality, and replaced the original sentence with “The small amounts of vinblastine and vincristine in C. roseus plants thus make the semi-synthetic route from the monomeric precursors (vindoline and catharanthine) the main commercial production method. Therefore, optimizing cultivation strategies is critical to increase the availability of pharmacologically valuable MIAs in C. roseus plants.”
- Literature review on fertilization (lines 49–64)
- The review reads as a fragmented list. Stronger logical connections are needed, for example: “Taken together, these studies indicate that nutrient supply can modulate MIA biosynthesis, although often at the expense of growth performance.”
- A concluding paragraph should emphasize the research gap: despite extensive studies on N, K, Ca, and Zn, few have investigated Mg, and none have explored the combined role of Mg and H₂, highlighting the novelty of the present study.
Response 2:
Thank you for your comments, We have added the sentence “Taken together, these studies indicate that nutrient supply can modulate MIA biosynthesis, although often at the expense of growth performance” after the literature review to strengthen the logical connection.
Thank you for your comments, We have revised the sentence as suggested, and replaced the original sentence with “Despite extensive studies on N, K, Ca, and Zn, few have investigated Mg application, and none have explored the combined role of Mg and H₂. This undiscovered territory constitutes the primary novelty of our work.”
- Section on magnesium (lines 65–76)
- Provides useful background but lacks a clear transition to C. roseus. References to potato and tobacco are informative but should be linked to the present study, for example: “These findings suggest that Mg fertilization can influence alkaloid metabolism in various plant species, yet its role in MIA production in C. roseus remains unclear.”
- If available, any studies on Mg effects specifically in C. roseus should be mentioned.
Response 3:
Thank you for your comments, We have revised and as suggested and mentioned one reference about Mg deficiency in C. roseus. The replaced sentence now reads: “These findings suggest that Mg fertilization can influence alkaloid metabolism in various plant species. It was reported that Mg deficiencies reduced ajmalicine concentration in C. roseus[11], yet its role in MIA production remains unclear.”
- Section on hydrogen (lines 77–89)
- The discussion on hydrogen is informative but overly technical; details on reaction kinetics (“extremely slow reaction kinetics…”) could be condensed or relocated to the Methods or Discussion section.
- More importantly, the introduction of hydrogen biology in plants is currently limited to seed germination and hormone regulation. For the present study, which aims to examine the effects of Mg + H₂ fertilization on MIA production in C. roseus, it is critical to include references to prior studies investigating the effects of hydrogen on secondary metabolites and/or on genes and enzymes involved in their biosynthetic pathways.
- Furthermore, if available, the authors should cite prior literature on the combined effects of Mg + H₂ on plant growth, development, and secondary metabolite accumulation in medicinal plants. This would provide a stronger rationale for the study and better situate it within the existing research context.
- If no such studies exist, this gap further highlights the novelty and relevance of the present investigation.
Response 4:
Thank you for your comments, We have condensed the sentence as suggested, and replaced the original sentence with “Upon hydrolysis, one molecule of MgH2 release two molecules of H2. Mg powder (MgP) has similar reaction in water, but produce one molecule of H2. Their ability to release H2 enables both MgH2 and MgP to act as combined Mg and H2 fertilizers.”
Thank you for your comments, We have revised and cited as sugestted. The modified sentence now reads: “Critically, emerging evidence suggests that molecular hydrogen can directly regulate the biosynthesis of secondary metabolites. For instance, treatment with hydrogen-rich water (HRW) is documented to significantly upregulate the phenylpropanoid biosynthesis pathway and increase the content of flavonoids and coumarins in the medicinal plant Ficus hirta[23].”
We have added the 23th reference “Zeng, J.; Yu, H. Integrated Metabolomic and Transcriptomic Analyses to Understand the Effects of Hydrogen Water on the Roots of Ficus hirta Vahl. Plants 2022, 11, 602. https://doi.org/10.3390/plants11050602”.
Thank you for your comments, We have revised and cited as sugestted. The modified sentence now reads: “Furthermore, recent evidence has revealed that the Mg-H₂ combination, supplied as MgH₂, can activate complex defense mechanisms in plants, as demonstrated by its ability to enhance cadmium tolerance in rice through hydrogen-mediated epigenetic regulation [27]. However, the potential of this synergy to influence the biosynthesis of valuable secondary metabolites in medicinal plants remains unexplored.”
We have added the 28th reference “Wang, P.; Lu, J.; Cao, J.; Wang, X. Exogenous MgH2-Derived Hydrogen Alleviates Cadmium Toxicity Through m6A RNA Methylation in Rice. Journal of Hazardous Materials 2024, 465, 133411. https://doi.org/10.1016/j.jhazmat.2024.136073.”
- Final paragraph / study objective (lines 90–93)
- The closing paragraph appropriately describes the study objectives, but it lacks a clearly stated hypothesis. Formulating an explicit hypothesis would improve clarity. For example: “We hypothesized that MgH₂ and Mg powder, by simultaneously supplying Mg and releasing H₂, would improve Mg uptake, plant growth, and MIA biosynthesis in C. roseus compared to conventional Mg fertilizers.”
- Additionally, adding a transition sentence linking the research gap to the study objective would strengthen the rationale. For example: “Given the limited information on hydrogen’s effects on secondary metabolites and the lack of studies on combined Mg + H₂ application in medicinal plants, it is important to evaluate their impact on C. roseus growth and MIA production.”
- These additions would make the final paragraph more explicit, logical, and readable, clearly connecting background, gap, and study aim.
Response 5:
Thank you for your comments, We have provided a clearly stated hypothesis as suggested “We hypothesized that MgH₂and Mg powder, by simultaneously supplying Mg and releasing H₂, would improve Mg uptake, plant growth, and MIA biosynthesis in C. roseus compared to conventional Mg fertilizers.”
Thank you for your comments. We have revised the sentence as suggested, and replaced the original sentence with “Given the limited information on hydrogen’s effects on secondary metabolites and the lack of studies on combined Mg + H₂ application in medicinal plants, it is important to evaluate their impact on C. roseus growth and MIA production.”
Overall Summary – Introduction
- Improve logical flow and connections between previous studies.
- Emphasize the research gap in Mg and Mg+H₂ fertilization.
- Reduce overly technical chemical details; include references on hydrogen’s effects on secondary metabolism.
- Formulate a clear working hypothesis and provide a smooth transition from the identified research gap to the study aim.
Response:
Thank you for your comments. We have improved logical flow and connections between previous studies, emphasized the research gap in Mg and Mg+H₂ fertilization, reduced overly technical chemical details, included references on hydrogen’s effects on secondary metabolism, formulated a clear working hypothesis and provided a smooth transition from the identified research gap to the study aim.
Comments on Materials and Methods
- Section 2.1 – Plant Materials and Treatments
- The description of the plant material and fertilizer treatments is generally clear. However, the rationale for selecting the four Mg fertilizers could be briefly stated in the Methods. In the Introduction, MgO and MgSO₄ were described as conventional slow- and rapid-release Mg sources, while MgH₂ and Mg powder were introduced as novel Mg + H₂ fertilizers. Explicitly restating this classification in Section 2.1 would improve coherence and help readers understand the comparative design of the study.
- The concentration of 7 mg/kg soil is provided, but the method for calculating or standardizing this dose across different fertilizer forms is not fully explained. Including a brief note on how equivalent Mg concentrations were achieved would improve reproducibility.
- While it is specified that 20 plantlets per treatment were collected for subsequent analyses, the number of pots or experimental units from which these plants were obtained is not indicated. Clarifying this point would strengthen transparency regarding biological replication and the independence of samples.
Response 1:
Thank you for your comments. We have restated the classification of fertilizers in Section 2.1 as “…, and germinated in the substrate soils treated with four forms of magnesium fertilizers, MgSO4, MgO, MgH2, and Mg powder (MgP) (50 μm,98%), representing conventional Mg rapid- or slow-release sources, and novel Mg + H₂ slow-release sources.” The concentration of 7 mg/kg soil is the Mg concentration. The actual application rate of each magnesium fertilizer was recalculated from this 7 mg Mg kg⁻¹ benchmark according to its molecular mass. We have added this information in Section 2.1. We transplanted 1 plantlet into 1 pot and we added this information in Section 2.1.
- Section 2.2 – Seedling Emergence
- The methodology is detailed, but the explanation of the t50% equation could be formatted more clearly. Some variables (e.g., ed - ed-1) are confusing as currently written.
- It would be useful to specify how and when emergence counts were performed (e.g., time of day) and whether observers were blinded to treatments to reduce potential bias.
- “The trays were arranged in a completely randomized design”à this information would be more appropriate in the Statistical Analysis section.
Response 2:
Thank you for your comments. For clarity, the t50% equation has been reformatted in Section 2.2 as “T50% = T + (50% - X)/ (Y - X), in which T50% is the time for emergence of 50% of the seeds, T is the day before 50% was reached, X is the emergence (%) observed in T, and Y is the emergence (%) in the day it was ≥50%.” The observers were blinded to the treatments. There were two students involved in the seedling emergence experiment. One was in charge of the experiment performance and the other was in charge of recording the data. The trays used in this experiment were the same specifications and only labeled by numbers (1, 2, 3, 4, 5). We have moved “The trays were arranged in a completely randomized design” to the Statistical Analysis section.
- Section 2.3 – Plant Height and Weight
- Methods are generally clear, but it is recommended to specify whether measurements were taken on fresh or dry weight (or both) for clarity and consistency. For example, in the Results, plant weight is reported as fresh weight, whereas chlorophyll, Mg, and alkaloid concentrations are measured on powdered leaf samples, with alkaloids expressed per dry weight. Clarifying the basis of all measurements would help readers correctly interpret and compare the data.
- The number of plantlets per replicate is given (five), but indicate whether these five were pooled for weight or measured individually, as this affects statistical analysis.
Response 3:
Thank you for your comments. Plant weight was fresh weight and the plantlets were measured in pool per replicate. The measurement of plant height and weight was performed firstly, then we ground leaves into powder and stored in -80℃ for other experiment. According to the requirement of each experiment, samples for MIA measurement were lyophilized for 72 hours, and alkaloid measurement were taken on dry weight (DW). The others used the fresh leaf powder samples of C. roseus plantlets. These information has been specified in the Section 2.3 to clarify the methods.
- Section 2.4 – Chlorophyll Analysis
- The extraction and measurement procedure is adequately described. However, “grinded” should be corrected to ground.
- The protocol mentions ELISA reader for absorbance measurement; typically, chlorophyll is measured by spectrophotometer. Clarify whether the ELISA reader was used as a plate reader for absorbance, and provide model specifics if possible.
- Methods report readings at 646 nm but the Arnon equations are written with A₆₄₅. Please state which absorbance was actually used for calculations (A₆₄₅ or A₆₄₆) and justify the interchange if applicable.
Response 4:
Thank you for your comments. “grinded” has been corrected to “ground”. We used microplate scanning spectrophotometer (PowerWave XS) to measure chlorophyII content and have corrected it with model specifics in the manuscript. The absorbance at 645 nm was used for calculations and we have corrected “646 nm” to “645 nm” in the Section 2.4.
- Section 2.5 – Mg Content Determination
- Overall method is clear. Recommend specifying the digestion vessel type (glass/teflon) and whether blanks and standards were included for ICP-OES calibration.
- Clarify the sample preparation step: “moistened with a small amount of water” is vague; provide exact volume or procedure for reproducibility.
Response 5:
Thank you for your comments. The digestion vessel type was teflon, and blanks and standards were included for ICP-OES calibration. We moistened with 5 mL water and we have corrected it in the Section 2.5.
- Section 2.6 – qRT-PCR
- Procedure is generally clear, but the storage temperature is given as -20 °C; for RNA stability, -80 °C is more typical. Confirm accuracy.
- Although primers are listed in Supplementary Table S1, indicate the amount of RNA used for cDNA synthesis to ensure reproducibility.
- Include details about PCR cycling conditions (temperatures, times, number of cycles) to enhance reproducibility.
Response 6:
Thank you for your comments. “-20 °C” has been corrected to “-80 °C”. We have clarified that 500 ng RNA was used for cDNA synthesis in the Section 2.6. PCR cycling conditions was listed in Table S2.
- Section 2.7 – MIAs Analysis
- The UPLC-Q/TOF-MS methodology is comprehensive and detailed.
- It would be helpful to briefly explain the rationale for the choice of column, gradient program, and PDA detector wavelengths, to clarify why these conditions were optimal for MIAs separation and quantification.
Response 7:
Thank you for your comments. The UPLC-Q/TOF MS column, gradient program and PDA wavelengths were selected following the optimized conditions reported by Pan et al. (2019). We have added the information in the section 2.7 and the reference in the manuscript.
- Section 2.8 – Soil pH Measurement
- Methods are generally adequate, but clarify at which time point(s) soil pH was measured and whether multiple readings were taken per mixed sample to ensure reproducibility.
Response 8:
Thank you for your comments. Soil pH was measured at 9 am. Three replicate readings were taken for each mixed sample. We have added the information in the Section 2.8.
- Section 2.9 – Statistical Analysis
- Use of ANOVA and Pearson correlation is appropriate.
- Recommend clarifying whether data were tested for normality and homogeneity of variance before ANOVA (specify the type of test used: graphical tests and/or analytical).
- Specify whether multiple comparison tests (e.g., Tukey’s HSD, Sidak, Fisher LSD etc) were performed after ANOVA to identify significant differences among treatments.
Response 9:
Thank you for your comments. Data were tested for homogeneity of variance before ANOVA with Levene test (Supplemental datasheet1). Duncan's multiple range test and Fisher LSD were performed after ANOVA to identify significant differences among treatments (Supplemental datasheet1). We have added the information in the Section 2.9.
Overall Summary - Materials and Methods
- Methods are generally well described, but some sections could benefit from clarification for reproducibility: e.g., Mg dose standardization, sample preparation details, RNA quantity, PCR cycling, and replication strategy.
- Minor corrections needed for language (e.g., “grinded” → ground) and consistency (wavelengths, notation).
- Consider adding more details on quality controls, standard curves, and statistical assumptions to strengthen rigor and reproducibility.
Response:
Thank you for your comments. We have made clarification for Mg dose standardization, sample preparation details, RNA quantity, PCR cycling, and replication strategy as suggested. Minor corrections have been made for language. We have added more details on quality controls, standard curves, and statistical assumptions.
Comments on Results-qxm
- Section 3.1 – Seed Germination
- The results are clearly presented, including both percentage emergence and T50% values.
- Minor suggestion: the text says improvements were “not significantly” – consider rephrasing as “not statistically significant” for clarity in academic writing.
Response 1:
Thank you for your comments. We have revised the sentence as suggested and already replaced “not significantly” with “no statistically significant”.
- Section 3.2 – Plant Growth
- The results are clearly presented, and reporting percentage increases is informative.
- The distinction in effects between MgH₂/MgP and MgO/MgSO₄ treatments is clear and well supported by the data.
Response 2:
Thank you for your comments.
- Section 3.3 – Chlorophyll Analysis
- The trends for chlorophyll a and b are described well, and percentage changes are provided.
- Minor language corrections: “grinded” → ground (if repeated here from Methods).
- Suggest clarifying the Results text by replacing ‘Mg’ with ‘MgP’ for consistency with the treatment names used in the Methods (MgSO₄, MgO, MgH₂, MgP), to avoid ambiguity regarding which magnesium treatment caused the observed effects on chlorophyll b.
- Suggest using precise statistical language when reporting comparisons. For example, replace ‘Mg and MgH2 treatments significantly enhanced chlorophyll b, while MgSO4 caused a sharp decline’ with ‘MgP and MgH2 produced a significantly higher chlorophyll b content compared to CK, MgO, and MgSO4, whereas MgSO4 significantly decreased chlorophyll b relative to MgP and MgH2.’ This statistically accurate language should be applied consistently throughout the entire Results section to clearly indicate which differences are significant.
Response 3:
Thank you for your comments. We have revised the sentence as suggested. Firstly, we replaced “grinded” with “ground”; secondly, we replaced “Mg” with “MgP”; last, we replaced “Mg and MgH2 treatments significantly enhanced chlorophyll b, while MgSO4 caused a sharp decline” with “MgP and MgH2 produced a significantly higher chlorophyll b content compared to CK, MgO, and MgSO4, whereas MgSO4 significantly decreased chlorophyll b relative to MgP and MgH2.”
- Section 3.4 – Magnesium Content
- Mg accumulation is clearly reported with percentage increases; the ranking of fertilizers is evident.
- Minor suggestion: replace “but both not significantly” with “though not statistically significant” for correct academic phrasing.
Response 4:
Thank you for your comments. We have revised the sentence as suggested and we have replaced “but both not significantly” with “though not statistically significant”.
- Section 3.5 – Soil pH
- Results are adequately reported, with numerical pH values.
- Suggest clarifying the phrasing: “a certain increase” → “a modest increase” or “an increase of X units”.
- Consider explicitly stating whether the MgH₂-induced increase (to 6.38) was statistically significant compared to the control.
- Fig. 3: Maintain consistency in naming treatments for all figures (Mg or MgP); same in the text
Response 5:
Thank you for your comments.We have revised the sentence as suggested and we have replaced “a certain increase”with“a modest increase”. Besides, we have stated that MgH₂-induced increase (to 6.38) was statistically significant compared to the control. We have replaced “Mg” with “MgP” in Fig 3 to maintain consistency in naming treatments.
- Section 3.6 – Gene Expression in MIA Pathway
- Very detailed reporting of gene expression changes. Percent increases over control are provided, which is excellent.
- Suggest minor rewording for readability: long lists of percentages could be summarized as “Most genes in the MIA pathway were significantly upregulated under MgH2 and MgP treatments, with maximum increases of 569% (Redox1) and 223% (TDC), respectively (Fig. 4).” Detailed values can remain in the figure legend or supplementary table.
- Consider consistency in gene naming (e.g., 16OMT vs T16OMT) and confirm all genes mentioned match Table S1.
Response 6:
Thank you for your comments. The minor suggestion to reword long list of percentages is greatly appreciated. To facilitate readers’ correct and easy comparison of the data, we have summarized part of long lists of percentages and remained them in Fig 4, but we preferred to retain part of the key list of percentages if this is not considered a major concern. The paragraph has been reworded as “Four fertilizers showed positive effect on most genes expression, especially the expression of vindoline and vinblastine biosynthetic genes (T16H, T3R, NMT, 16OMT, D4H, DAT and PRX1) (Fig 4). MgH2 treatment significantly induced 14 genes expression and reduced 2 genes expression. MgP treatment led to significant increase in the expression of 13 genes but significant decrease in 4 genes expression. MgO treatment significantly upregulated 12 genes expression and downregulated 2 genes expression. The application of MgSO4 resulted in significant increase in 10 genes expression, but significantly reduced 6 genes expression. Under the application of four fertilizers, 12 genes showed the highest expression in MgH2-treated group, including TDC, GS, CS, TS, T16H, T3R, NMT, D4H, DAT, PRX1, CrEIN3 and ORCA3 with maximum increase of 223%, 7%, 569%, 53%, 94%, 180%, 276%, 119%, 194%, 160%, 58% and 254% over the control, respectively. Redox1 showed the highest expression in MgP-treated group with an increase of 55%. SGD showed the highest expression in MgO-treated group with an increase of 24%. STR, PAS, 16OMT and CrWRKY1 had the highest expression in MgSO4-treated group with increase of 39%, 34%, 250%, and 64%, while SGD, GS, GO, Redox1, DPAS and TS had the lowest expression with significant decrease of 38%, 26%, 50%, 22%, 60% and 48% over the control, respectively.” At last, we have confirmed all genes metioned match Table S1.
- Section 3.7 – Alkaloids Analysis
- The results are clearly presented, with fold changes and percentage increases that highlight treatment effects.
- Suggest minor stylistic adjustments: instead of “And MgO treatment increased…”, use “MgO treatment also increased…”
- The comparison of MgSO4 with other treatments is clear and highlights the contrasting effect; well presented.
- Additional suggestion: for consistency across the Results section, consider reporting changes either uniformly as percentages or as fold changes, rather than mixing both formats. This would improve clarity and readability for the audience.
Response 7:
Thank you for your comments. We have revised the sentence as suggested and replaced“And MgO treatment increased…”with“MgO treatment also increased…”.Besides, we have reported all the changes as percentages to improve clarity and readability for the audience.
- Section 3.8 – Correlation Analysis
- Correlation analyses are well described. Positive and negative correlations are clearly reported with significance levels.
- Suggest clarifying some phrasing: “extremely significantly positively correlated” → “highly significant positive correlations (p < 0.001)” for standard academic style.
- Consider grouping correlation results by category for clarity (e.g., growth traits, chlorophyll/Mg, gene expression, MIAs).
- The link between gene expression, plant traits, and H2 release is well presented; possibly add a brief summary sentence highlighting the main findings from Figures 6–7.
Response 8:
Thank you for your comments. We have revised the sentence as suggested and we have replaced“extremely significantly positively correlated”with “highly significant positive correlations (p < 0.001)”. We have grouped the correlation results by p value or category (gene expression, MIAs) and added a brief summary sentence highlighting the main finding. We have added a brief summary sentence as “Overall, the correlation analysis revealed that MIA biosynthesis in C. roseus was significantly associated with H₂ release, soil pH, Mg uptake and chlorophyll content.” To highlight the main findings from Figures 6-7.
- Additional suggestion – ANOVA results table
it would be highly beneficial for transparency and reproducibility if the authors provide a summary table of the ANOVA results for all measured variables.
The table could include:
- F-values for each variable,
- Corresponding p-values,
- Symbols (e.g., *, **, ***) for different significance levels, and
- “ns” for non-significant comparisons.
This table could be included either in the main text (if concise) or in the Supplementary Materials to avoid overloading the Results section. Providing such a table would enhance clarity, allow readers to quickly assess the statistical robustness, and strengthen the overall presentation of the study.
Response 9:
Thank you for your comments. We have provided the table (SI datasheet1) in the Supplementary Materials.
- Additional suggestion – Figure legends
- It is recommended that the authors specify in each figure legend the type of multiple comparison test used (e.g., Tukey’s HSD, LSD) and the sample size for each treatment (n = …). Including this information would improve clarity and reproducibility, allowing readers to fully understand the statistical analyses underlying the reported significant differences.
Response 10:
Thank you for your comments. We have specified in each figure legend the type of multiple comparison test used and the sample size for each treatment (n = …).
- Additional suggestion – Consistency in style and significance reporting
- Ensure consistency in the notation of chemical formulas (e.g., MgH₂ vs MgH2) and abbreviations (SE, TFE, MIAs).
Response 11:
We have ensured consistency in the notation of chemical formulas (e.g., MgH₂ vs MgH2) and abbreviations (SE, TFE, MIAs). TFE has been replaced with T50%.
Overall Summary – Results (revised)
- Results are robust, detailed, and generally well-structured.
- Ensure consistent naming of treatments (MgP, MgO, MgH₂, MgSO₄) throughout text, figures, and tables.
- Use precise statistical language when reporting comparisons, indicating which differences are significant.
- Provide an ANOVA summary table for transparency.
- Clarify figure legends: specify sample size (n) and multiple comparison tests used.
- Rephrase ambiguous or informal phrases (e.g., “grinded” → “ground”, “a certain increase” → “modest increase of X units”).
- Summarize extensive gene expression and alkaloid results for readability, while keeping detailed values in figure legends or supplementary tables.
- Ensure consistency in units, abbreviations, and notation across the Results section.
- Maintain this precise and statistically accurate language consistently throughout the Results.
Response:
Thank you for your comments. As suggested, we have kept consistent naming of treatments, used precise statistical language, provided an ANOVA summary table for transparency, clarified figure legends, rephrased ambiguous or informal phrases, summarized part of extensive gene expression for readability (detailed values in Fig 4), ensured consistency in units, abbreviations, and notation across the Results section, and maintained this precise and statistically accurate language consistently throughout the Results.
Comments on Discussion
- General structure and clarity:
The discussion remains generally well-organized and logically links the results to previous studies on Mg and H₂ fertilization. Some sentences are still long and could be split for clarity. Statements largely distinguish between results and interpretations now, but minor adjustments could further separate reporting from mechanistic explanation. Consider also maintaining consistent terminology (e.g., MgH₂ vs MgH2, MIA vs MIAs).
Response 1:
Thank you for your comments. We have made minor adjustments as adding “Moreover”, “In contrast”, “we found that”, “It is notable that”, “Based on these findings, we propose...”, “These results suggest that...” to clearly distinguish results report and mechanism interpretation. Lastly, we have standardized the chemical nomenclature throughout the manuscript, using MgH₂and MIA consistently.
- Integration with literature:
References on Mg effects are adequate, and H₂ literature is cited. However, references on hydrogen’s role specifically in secondary metabolite biosynthesis or regulation of biosynthetic genes in plants (especially in roseus or medicinal plants) are still sparse. Adding these would strengthen the rationale and highlight the novelty of the Mg + H₂ combination.
Response 2:
Thank you for your comments, We have added “Futhermore, H2 treatment directly upregulated key structural genes in the phenylpropanoid biosynthetic pathway and enhanced the accumulation of flavonoids and coumarins in the medicinal plant Ficus hirta[23]” in the end of the 4th paragraph.
- Mechanistic explanations:
The MIA biosynthetic pathway discussion is thorough. Repetitive details on Mg’s role in chlorophyll/light absorption can be condensed or integrated more smoothly with gene expression and alkaloid results to improve readability.
Response 3:
Thank you for your comments. We have condensed the original sentence as “The role of Mg in photosynthesis is well-established as the central atom of chlorophyll.” To integrated smoothly with gene expression and alkaloid results for readability, we have added a sentence “Our results revealed that the expression of most MIA biosynthetic genes, especially those in the vindoline pathway, exhibited positive correlations with Mg and chlorophyll contents in C. roseus leaves.” in the 4th paragraph of the discussion.
- Statistical and quantitative clarity:
Quantitative data and statistical significance have already been appropriately presented in the Results. In the Discussion, ensure statements such as “positive effect” or “better absorption” are explicitly linked to the relevant figures/tables rather than repeating numbers unnecessarily.
Response 4:
Thank you for your comments. “positive effect” has been explicitly linked to Fig 1 and “better absorption” has been explicitly linked to Fig 4. We have simplified the unnecessary numbers, and replaced the original sentence with “This aligns with previous reports that slow-release Mg fertilizers can improve crop yield more effectively than rapid-release alternatives.”
- Figures and interpretation:
Figure 8 is now properly referenced in the text when interpreting results, so this point has been addressed.
Response 5:
Thank you for your comments.
- Transitions and flow:
Transitions could still be slightly improved. For instance, after discussing Mg effects, a short sentence highlighting the rationale for adding H₂ would make the link to Mg + H₂ effects more explicit.
Response 6:
Thank you for your comments. We have added the sentence to transit “Given the reported roles of Mg and H2 in influencing plant secondary metabolism, we sought to determine whether their combination would yield synergistic effects on MIA biosynthesis” in the last paragraph of the discussion.
- Language and style:
Minor language edits are suggested: e.g., “had slightly effect” → “had a slight effect”; “similar as gene expression” → “similar to gene expression.” Avoid repetitive phrases like “significantly positive correlation” repeated multiple times.
Response 7:
Thank you for your comments. We have revised as suggested, and replaced the original sentence with “In contrast, MgSO4 had a slight effect on both plant weight and height of C. roseus.” and “This change of alkaloid content was similar to gene expression.” The repetitive phrases like “significantly positive correlation” have been modified as “positively correlated” and appeared only once in the discussion.
Overall Summary – Discussion
- Well-organized, logically links results to literature.
- Include references on H₂ effects on secondary metabolism and gene regulation in medicinal plants, ideally including C. roseus.
- Strengthen transitions between discussion of Mg alone, H₂ alone, and Mg + H₂ combination.
- Condense repetitive mechanistic explanations to improve readability.
- Ensure all qualitative statements are clearly linked to figures/tables.
- Minor language adjustments recommended.
Response:
Thank you for your comments. We have logically linked results to literature, included references on H₂ effects on secondary metabolism and gene regulation in medicinal plants, strengthened transitions, condensed repetitive mechanistic explanations, linked qualitative statements to figures, and adjusted minor language problems.
Comments on Conclusions
Content clarity:
The conclusions accurately summarize the main findings: Mg + H₂ fertilizers (MgH₂ and MgP) outperform Mg-alone fertilizers in promoting Mg uptake, plant growth, and MIA production.
Practical implications:
The potential application for commercial cultivation is clearly stated, reinforcing applied relevance.
Language and phrasing:
Minor improvements: e.g., “These findings have practical guidance to commercial production” → “These findings provide practical guidance for commercial production.” Sentences could be made slightly more concise and avoid repeating mechanistic details.
Future directions and limitations: Conclusions lack future research directions or limitations. I recomende to include them.
Overall Summary – Conclusions
- Concisely summarize findings.
- Emphasize novelty and practical impact of Mg + H₂ fertilization.
- Minor language improvements for clarity.
- Add a brief note on future research directions or limitations to frame the study within a broader context.
Response:
Thank you for your comments. We have summarized the main findings, stated the potential application for commercial cultivation and reinforced applied relevance. We have replaced the sentence with“These findings provide practical guidance for commercial production.”Besides, we have emphasized novelty and practical impact of Mg + H₂fertilization and added future research directions or limitations.

Reviewer 3 Report
Comments and Suggestions for Authors
Dear Authors,
Suggestions for corrections are in the attached file.
Kind regards,

Author Response
Reply to reviewer 3
Reviewer3:
Dear Authors, I appreciated your approach to demonstrating the synergistic effects of magnesium and hydrogen fertilization on the growth and alkaloid biosynthesis of Catharanthus roseus, focusing on modulating monoterpenoid production. The proposed revision suggestions may be accepted or rejected. If not, please provide a justification.
Reply:
We sincerely thank you for the positive feedback on our work and for providing these constructive suggestions. We have carefully addressed each point raised, as detailed below. The corresponding revisions have been made in the manuscript.
- a) Dear Authors, if possible, please indicate the main results with quantitative data, this will allow us to connect the discussion in the summary with what was presented in Figures 4 and 6.
Response to Comment a):
We agree with the reviewer that providing quantitative data in the summary enhances clarity. Accordingly, we have revised the Abstract to include key quantitative findings and explicitly link them to the relevant figures.
Specific changes made: In the Results section of the Abstract, we now state: "F MgH₂ and MgP fertilization significantly increased plant weight by 60% and 29% over the control, respectively." We further specify: "Among them, MgH₂ yielded the highest contents of catharanthine, vindoline and ajmalicine, reaching 1.67-, 1.49- and 5.17-fold of the control, respectively. " Finally, we have linked the correlation analysis to its figure: "Pearson correlation analysis showed significant positive correlations among H₂ release, soil pH, and Mg uptake, as well as with plant growth and MIA content." This revision directly connects the summarized discussion to the quantitative data presented in both Figures 4 and 6, as requested.
- b) See if it is possible to specify the interaction between magnesium and hydrogen in optimizing the evaluated parameters.
Response to Comment b):
We thank the reviewer for this insightful suggestion. We agree that elaborating on the Mg-H₂ interaction is crucial and have revised the Discussion section to specify this potential synergy more clearly.
Specific changes made: We have added a dedicated sentence and expanded the reasoning in the Discussion: "These results suggest a synergistic interaction between magnesium and hydrogen. These fertilizers not only supply Mg²⁺ but also release H₂ during hydrolysis. We propose that the released H₂, potentially by modulating plant signaling pathways or improving the rhizosphere environment, enhances the plant's efficiency in magnesium uptake and utilization. This synergy ultimately led to the observed improvements in chlorophyll synthesis, plant biomass, and the upregulation of MIA biosynthetic genes, culminating in higher alkaloid accumulation."
- c) What is the practical applicability of the study in the production of secondary metabolites?
Response to Comment c):
We appreciate the reviewer's question regarding practical applicability. We have strengthened the Conclusion section to better highlight the practical implications and potential applications of our findings.
Specific changes made: The Conclusion now more explicitly states: "These findings provide practical guidance for commercial production to enhance the yield of valuable anti-cancer alkaloids. Furthermore, this study lays the groundwork for future research to explore the application of Mg plus H₂ fertilization in other medicinal plants under field conditions."
Once again, we extend our gratitude to you for your valuable time and insightful comments. We believe the manuscript has been substantially improved by addressing these points and hope it now meets the high standards required for publication.

Round 2
Reviewer 2 Report
Comments and Suggestions for Authors
The Authors have addressed most of the revision requests satisfactorily. However, a few points still require clarification and/or correction. Once these issues are addressed, I would consider the manuscript acceptable for publication:
- Italicization of species name: Ensure that C. roseus is consistently italicized throughout the manuscript (e.g., in the Introduction).
- Font and formatting issues: Correct the font size/character problem in sentences such as “Samples for MIA measurement were lyophilized for 72 hours” and “but not statistically significant (Table 1),” and check for other similar occurrences throughout the text.
- Potential inconsistency regarding post-hoc tests: In the main text (Section 2.9, Statistical Analysis), the authors report using Duncan’s test following ANOVA to detect significant differences among treatments, whereas in the supplementary data the same variables (plant height, fresh weight, magnesium content, chlorophyll content, soil pH, and vindoline, catharanthine, and ajmalicine contents) are analyzed using Fisher’s LSD test. From the supplementary tables, it appears that comparisons were conducted only between each treatment and the control, without comparisons among the treatments themselves. Fisher’s LSD is typically used for all pairwise comparisons, i.e., to compare all possible pairs of treatments, and only for a limited number of groups (typically ≤3, as the risk of Type I errors increases with more comparisons due to the lack of correction). It is not appropriate for multiple comparisons against a reference group (control); the correct test in this context would be the Dunnett test. It is therefore recommended to clarify in Section 2.9 which post-hoc test was actually used to derive the results reported as “significantly higher” or “no significant differences” compared with the control. If Fisher’s LSD was used for treatment vs. control comparisons in both the results and supplementary data, the authors should repeat the analysis using the Dunnett test and update the supplementary tables, as well as Section 2.9, to ensure transparency and statistical rigor.
Author Response
The Authors have addressed most of the revision requests satisfactorily. However, a few points still require clarification and/or correction. Once these issues are addressed, I would consider the manuscript acceptable for publication:
- Italicization of species name: Ensure that C. roseus is consistently italicized throughout the manuscript (e.g., in the Introduction).
Response 1:
Thank you for your comments. We have ensured that C. roseus is consistently italicized throughout the manuscript.
2. Font and formatting issues: Correct the font size/character problem in sentences such as “Samples for MIA measurement were lyophilized for 72 hours” and “but not statistically significant (Table 1),” and check for other similar occurrences throughout the text.
Response 2:
Thank you for your comments. We have corrected the font size problem in the manuscript.
3. Potential inconsistency regarding post-hoc tests: In the main text (Section 2.9, Statistical Analysis), the authors report using Duncan’s test following ANOVA to detect significant differences among treatments, whereas in the supplementary data the same variables (plant height, fresh weight, magnesium content, chlorophyll content, soil pH, and vindoline, catharanthine, and ajmalicine contents) are analyzed using Fisher’s LSD test. From the supplementary tables, it appears that comparisons were conducted only between each treatment and the control, without comparisons among the treatments themselves. Fisher’s LSD is typically used for all pairwise comparisons, i.e., to compare all possible pairs of treatments, and only for a limited number of groups (typically ≤3, as the risk of Type I errors increases with more comparisons due to the lack of correction). It is not appropriate for multiple comparisons against a reference group (control); the correct test in this context would be the Dunnett test. It is therefore recommended to clarify in Section 2.9 which post-hoc test was actually used to derive the results reported as “significantly higher” or “no significant differences” compared with the control. If Fisher’s LSD was used for treatment vs. control comparisons in both the results and supplementary data, the authors should repeat the analysis using the Dunnett test and update the supplementary tables, as well as Section 2.9, to ensure transparency and statistical rigor.
Response 3:
Thank you for your comments. We tested homogeneity of variance with Levene’s test prior to ANOVA, and the results indicated that all data met the assumption of equal variances. We therefore conducted Duncan’s test after ANOVA to identify significant differences among treatments; this test is appropriate for multiple pairwise comparisons when variances are homogeneous. As recommended, we have now also performed Dunnett’s test and reported the results in SI Datasheet 1. In addition, we have removed any reference to Fisher’s LSD test from Section 2.9.
